# QiMeng-ChipV-RTL: Exploiting Information Locality for IP-level Verilog Generation

**Hanqi Lyu** [1][2]  **Di Huang** [2][3]  **Yaoyu Zhu** [2]  **Kangcheng Liu** [2][3]  **Bohan Dou** [1][2]  **Chongxiao Li** [2][3]  **Pengwei Jin** [2]
**Shuyao Cheng** [2]  **Rui Zhang** [2][3]  **Zidong Du** [2][3]  **Qi Guo** [2][3]  **Xing Hu** [2][3]  **Yunji Chen** [2][3]

## Abstract

The generation of Register-Transfer Level (RTL) code is a crucial yet labor-intensive step in digital hardware design, traditionally requiring engineers to manually translate complex specifications into thousands of lines of synthesizable Hardware Description Language (HDL) code. While Large Language Models (LLMs) have shown promise in automating this process, existing approaches—including fine-tuned domain-specific models and advanced agent-based systems—struggle to scale to industrial IP-level design tasks. We identify three key challenges: (1) handling long, highly detailed documents, where critical interface constraints become buried in unrelated submodule descriptions; (2) generating long RTL code, where both syntactic and semantic correctness degrade sharply with increasing output length; and (3) navigating the complex debugging cycles required for functional verification through simulation and waveform analysis. To overcome these challenges, we propose *ChipV-RTL*, a multi-agent framework that leverages *information locality* in modular hardware design. ChipV-RTL decomposes the long-document to long-code generation problem into a set of short-document, short-code tasks, enabling scalable generation and debugging. Specifically, ChipV-RTL integrates hierarchical document partitioning, task planning, localized code generation, interface-consistent merging, and AST-guided locality-aware debugging. Experiments on REALBENCH, an IP-level Verilog generation benchmark, demonstrate that ChipV-RTL substantially outperforms state-of-the-art (SOTA) LLMs

and agents, achieving a pass rate of 45.0% compared to 21.6%. Code, project page are available at: https://iprc-dip.github.io/ChipV-RTL/.

## 1 Introduction

The generation of Register-Transfer Level (RTL) code is a core step in digital hardware design. This process is notoriously labor-intensive and error-prone, as engineers must manually translate natural language specifications into thousands of lines of synthesizable Hardware Description Language (HDL) code (e.g., Verilog, VHDL). The promise of Large Language Models (LLMs) to automate this step has spurred rapid innovation. Initial efforts focused on benchmarking general-purpose models (Liu et al., 2023b; Thakur et al., 2023) and developing domain-specific solutions through fine-tuning or data augmentation (Liu et al., 2024c; Cui et al., 2024; Liu et al., 2024b; Zhao et al., 2025). More recently, the field has shifted towards sophisticated agent-based systems that mimic human design workflows. Agents such as VerilogCoder (Ho et al., 2025) and MAGE (Zhao et al., 2024) operate autonomously for planning and debugging on complex Verilog generation problems.

Despite strong results on academic benchmarks like VerilogEval (Liu et al., 2023b), *a clear gap appears when applying current LLM-based methods to industrial hardware design.* This is particularly evident with REALBENCH (Jin et al., 2025), an IP-level benchmark derived from real-world open-source IP, which features significantly longer documentation (197.3 vs. 5.7 lines) and code lengths (241.2 vs. 15.8 lines) compared to VerilogEval. Directly using SOTA models or agents often leads to a sharp drop in performance, with many outputs failing to be even syntactically correct. This gap highlights a mismatch between current model capabilities and the high requirements of real-world hardware engineering, from which we observe three main challenges:

**Long-Document Handling.** IP-level specifications are typically verbose, featuring numerous I/O signals and submodules. While modern LLMs support 32k+ token windows, their functional accuracy diminishes as document complex-

[1]University of Science and Technology of China [2]SKL of Processors, Institute of Computing Technology, CAS [3]University of Chinese Academy of Sciences. Correspondence to: Yunji Chen <cyj@ict.ac.cn>.

*Proceedings of the 43rd International Conference on Machine Learning*, Seoul, South Korea. PMLR 306, 2026. Copyright 2026 by the author(s).

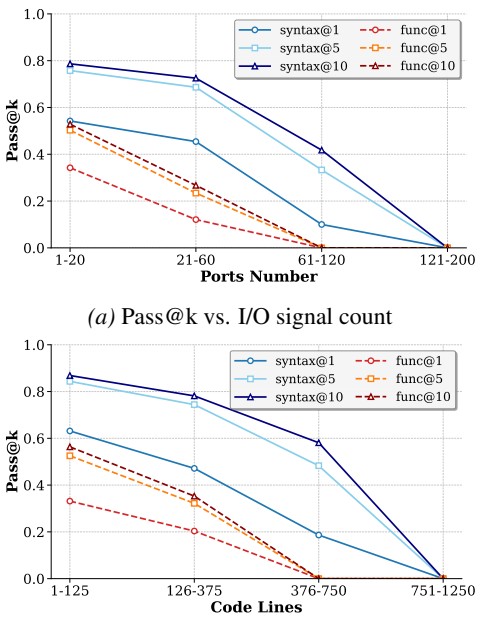

*(a)* Pass@k vs. I/O signal count

*(b)* Pass@k vs. code length

*Figure 1.* Performance of Claude 3.7 Sonnet on REALBENCH: Pass@k vs. (a) I/O signal count and (b) code length (lines), reporting syntactic and functional Pass@k. Accuracy decreases with interface complexity and output length.

ity grows. Overwhelmed by accumulating signal and module details, LLMs often miss critical interface constraints, resulting in "phantom" signals, port mismatches, and logic errors. As shown in Figure 1a, accuracy correlates negatively with I/O signal count.

**Long-Code Generation.** Increasing code length exacerbates the inherent weaknesses of LLMs in HDL code generation. As shown in Figure 1b, both syntactic and semantic accuracy drop significantly with code length. Beyond 750 lines, even $10\times$ repeated sampling rarely yields syntactically valid code. Common failures include incorrect macro references, use of non-synthesizable constructs, and fundamental syntax errors, underscoring the model's inherent limitations in generating reliable RTL code.

**Complex Debugging Process.** Although debugging has been addressed by prior frameworks, IP-level failures are harder to localize because they often arise from interactions among long specifications, numerous interfaces, and generated submodules. IP-level Verification relies on carefully constructed testbenches to ensure specification compliance. Each simulation failure triggers a laborious debugging cycle: engineers analyze waveforms to identify faulty signals, trace errors back to ambiguous or misinterpreted specification segments, and iteratively refine the design. This process not only corrects the code but also clarifies ambiguities in the specification itself, using waveform behavior as a definitive reference for refinement.

To address these challenges, we propose ChipV-RTL, a multi-agent framework explicitly designed for the real-world IP-level "long-document, long-code" hardware generation problem. Our key observation is that IP-level specifications inherit strong *information locality* from modular hardware design: code fragments can often be generated correctly by relying on only a portion of the document. This suggests that long-document to long-code generation can be decomposed into a set of short-document to short-code tasks without information loss, thereby mitigating the core challenges.

Specifically, ChipV-RTL organizes the following workflow as shown in Figure 2: (1) Preprocessing. Documents are partitioned into fragments with hierarchical indices. (2) Planning. Code structure is planned as sub-tasks with assigned document fragments. (3) Generation. Coding agents execute "short-document, short-code" generation for each sub-task. (4) Merging. Fragments are merged into a complete design with interface consistency. (5) Debugging. Error messages and AST-guided waveform analysis trace failures back to specification fragments for locality-aware debugging.

Our contributions are summarized as follows: (1) We identify the fundamental challenges hindering IP-level Verilog generation, namely, long-document handling, long-code generation, and the complex debugging process. (2) We identify the *information locality hypothesis*, positing that correct implementation for specific hardware modules is highly correlated with localized segments of the document rather than the global context. (3) Guided by this hypothesis, we introduce ChipV-RTL, an IP-level Verilog generation framework consisting of an index-driven document partitioning mechanism, fragment-based generation, and a traceable debugging pipeline that aligns error signals with relevant specification fragments via AST-guided analysis. (4) We conduct extensive experiments and analyses on realistic and challenging Verilog generation benchmarks, where ChipV-RTL achieves a 45.0% pass rate on REALBENCH (Jin et al., 2025) and 61.50% on CVDP cid003 (Pinckney et al., 2025), surpassing SOTA methods by 23.4% and 12.78%, respectively.

## 2 Related Work

**Benchmarks.** LLM-based RTL generation has emerged as a promising research area in electronic design automation (EDA). Foundational benchmarks such as VerilogEval (Liu et al., 2023b) and RTLLM (Lu et al., 2024) were established to systematically assess model performance, revealing both potential and limitations of off-the-shelf models. Reflecting industrial demands, recent benchmarks like RealBench (Jin et al., 2025) and CVDP (Pinckney et al., 2025) introduce significantly higher complexity. RealBench focuses on long-

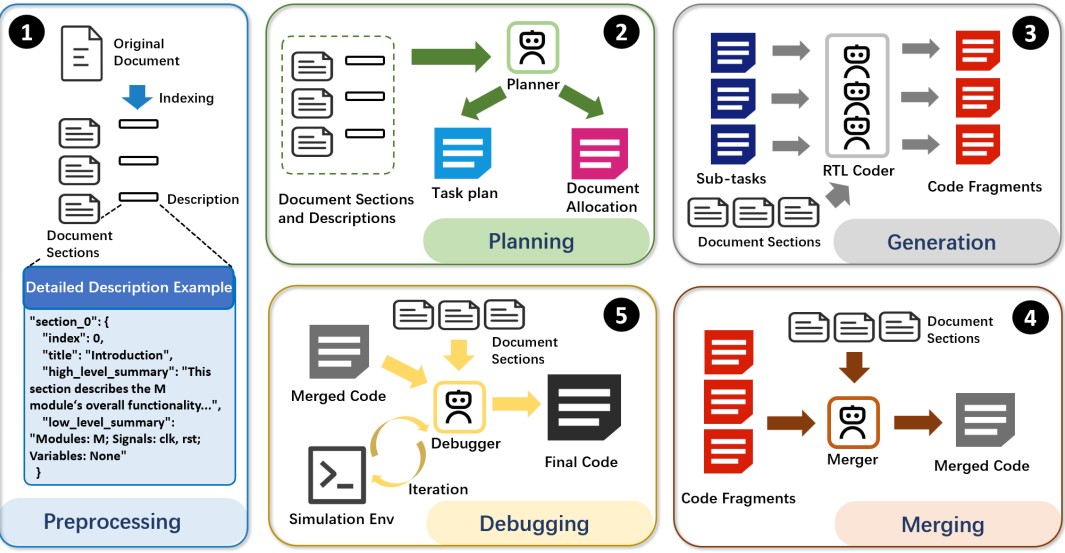

*Figure 2.* Workflow overview of ChipV-RTL.

context specifications and implementation, while CVDP covers generation, debugging, and optimization.

**LLM-based RTL Generation.** Initial studies (Nair et al., 2023; Blocklove et al., 2023) leveraged general-purpose LLMs for translating natural language specifications into HDLs like Verilog and VHDL, while subsequent efforts focused on domain-specific fine-tuning (Liu et al., 2024c; Thakur et al., 2024; Liu et al., 2023a; 2025a;b; Pei et al., 2024) and reinforcement learning (Zhu et al., 2025; Chen et al., 2025). While these approaches have shown promising progress, their evaluations are mostly centered on small-scale tasks. As a result, their effectiveness and scalability on complex, IP-level specifications remain insufficiently understood.

**Agent-based Frameworks for Hardware Design.** To overcome the limitations of single-pass generation, the field is shifting toward multi-agent frameworks that emulate iterative human workflows. For instance, MAGE (Zhao et al., 2024) employs a four-agent team—responsible for RTL generation, testbench creation, evaluation, and debugging—to establish a recursive design-refinement loop. Similarly, RTLSquad (Wang et al., 2025a) organizes agents into specialized squads for distinct project phases, namely exploration, implementation, and verification. Central to these systems is a task decomposition phase, where high-level specifications are partitioned into manageable sub-tasks to guide agents in coding and reflection, as seen in Spec2RTL-Agent (Yu et al., 2025) and VerilogCoder (Ho et al., 2025). Despite these advancements, autonomous IP-level generation remains hindered by three critical challenges which we aim to address: managing long-context

*Table 1.* Architectural comparison with prior frameworks.

| Component | ChipV-RTL | VerilogCoder | MAGE | Spec2RTL | RTLSquad |
|---|---|---|---|---|---|
| Doc preprocessing | ✓ locality-aware | × | × | ✓ summaries | × |
| Task decomposition | ✓ | ✓ | × | ✓ | ✓ |
| RTL generation | ✓ | ✓ | ✓ | △ C++ → HLS | ✓ |
| Iterative refinement | ✓ | ✓ | ✓ | ✓ | ✓ |
| Spec grounding | ✓ locality-aware | × | × | △ refined context | × |
| Human intervention | × | × | × | ✓ | × |

documentation, maintaining large-scale code coherence, and navigating complex debugging processes.

**Comparison with Prior Works.** We present a comparative analysis against representative Verilog generation frameworks, including MAGE, VerilogCoder, Spec2RTL, and RTLSquad. As summarized in Table 1, ChipV-RTL introduces critical innovations in document preprocessing and specification grounding based on the information locality hypothesis, designed to address the challenges of IP-level RTL generation from long-context specifications.

## 3 Methodology

We begin by fomulating the problem of IP-level Verilog generation (§3.1). We then introduce our key insight, the *information locality* in IP-level hardware specifications, with a quantitative analysis (§3.2), and finally present the ChipV-RTL framework built on this (§3.3).

### 3.1 Problem Formulation

We formulate the problem as follows:

**Input:** A natural language specification document $\mathcal{D}$, represented as an ordered sequence of $N$ semantic textual units

(e.g., paragraphs or sections), $\mathcal{D} = \{d_1, d_2, \ldots, d_N\}$. Also, a target module name $m$ and a simulation environment $E$ that provides golden execution feedback (including error messages and behavioral mismatches) for debugging purposes are given.

**Output:** A Verilog module $\mathcal{V}_m$. We model the generated code as a structured set of $M$ semantic code units instead of a monolithic text file, $\mathcal{V}_m = \{c_1, c_2, \ldots, c_M\}$. A code unit $c_j$ represents a functionally cohesive and syntactically complete block of RTL code, such as a module or a statement. The final output file is the concatenation of these units.

**Objective:** The generated module $\mathcal{V}_m$ must be functionally correct and can pass a suite of simulation tests from $E$ against a golden reference testbench, ensuring functional correctness.

## 3.2 Information Locality

Our approach is grounded in a core assumption we term ***information locality***: for any semantic code unit $c_j \in \mathcal{V}_m$, the information required to generate $c_j$ is primarily concentrated within a subset of the specification $\mathcal{D}$. This locality arises directly from the hierarchical and modular nature of hardware design. Complex systems are built from well-defined submodules (e.g., ALUs, register files), and IP-level specifications explicitly mirror this structure: dedicated sections describe each module's behavior, I/O, and internal logic. This creates a natural alignment, where the implementation of a code unit $c_j$ depends predominantly on its corresponding documentation segment. In contrast, general-purpose software specifications often describe high-level algorithms that do not decompose neatly into code-level constructs, leading to more diffuse information sources (Figure 3).

We quantify information locality by measuring the entropy of the information source distribution for each code unit. Our analysis begins by segmenting the specification $\mathcal{D}$ into paragraphs $\{d_i\}_{i=1}^N$ and the Verilog code $\mathcal{V}_m$ into statements $\{c_j\}_{j=1}^M$. For each code statement $c_j$, we utilize the conditional generation probability of LLMs to measure its relevance $S_{i,j}$ to every specification paragraph $d_i$. Here, $S_{i,j}$ is defined as the average log-probability of generating the tokens of code unit $c_j$ given the text segment $d_i$:

$$S_{i,j} = \frac{1}{|c_j|} \sum_{k=1}^{|c_j|} \log P(t_k \mid d_i, t_{<k}). \quad (1)$$

To ensure robust probability estimation across different scales, we apply Z-score normalization to the relevance scores of each code unit. Let $\mu_j$ and $\sigma_j$ be the mean and standard deviation of $\{S_{i,j}\}_{i=1}^N$. The normalized scores $\hat{S}_{i,j} = (S_{i,j} - \mu_j)/\sigma_j$ are then transformed into a conditional probability distribution $P(d_i \mid c_j)$ using a softmax

function with temperature $\tau = 0.1$:

$$P(d_i \mid c_j) = \frac{\exp(\hat{S}_{i,j}/\tau)}{\sum_{k=1}^N \exp(\hat{S}_{k,j}/\tau)}. \quad (2)$$

The locality for $c_j$ is then assessed by the entropy of this distribution:

$$H(c_j) = - \sum_{i=1}^N P(d_i \mid c_j) \log_2 P(d_i \mid c_j), \quad (3)$$

where lower entropy indicates that information is concentrated in a small number of textual units, thus supporting the locality hypothesis. To ensure comparability across specifications of different lengths, we normalize the entropy by its theoretical maximum, $H_{\max} = \log_2 N$, which occurs under a uniform distribution:

$$H_{\text{norm}}(c_j) = \frac{H(c_j)}{\log_2 N}. \quad (4)$$

This yields a scale-invariant measure. Finally, we average the normalized entropy across all $M$ code units:

$$\bar{H}_{\text{norm}} = \frac{1}{M} \sum_{j=1}^M H_{\text{norm}}(c_j). \quad (5)$$

A lower $\bar{H}_{\text{norm}} \in [0, 1]$ indicates stronger overall locality, which is comparable across varying $N$ and $M$.

We evaluate three tasks on information locality: (a) a **synthetic Verilog benchmark** (10 concatenated and renamed VerilogEval cases) as an ideal locality baseline (lower bound); (b) the **hardware IP** e203_cpu_top from REALBENCH; and (c) a **software counterpart** (LeetCode "Parse Lisp Expression" in Python) with comparable length. Row-normalized heatmaps and the average normalized entropy $\bar{H}_{\text{norm}}$ quantify locality strength. As shown in Figure 3, the hardware design (b) exhibits strong locality ($\bar{H}_{\text{norm}} = 0.6718$), much closer to the ideal (a) ($\bar{H}_{\text{norm}} = 0.6366$) than the software case (c) ($\bar{H}_{\text{norm}} = 0.8220$). This pattern holds across REALBENCH, where the average $\bar{H}_{\text{norm}} = 0.7261$ confirms strong locality in hardware specifications. To show the generalizability of these findings, we compute $\bar{H}_{\text{norm}}$ with various LLMs, including Qwen3-8B (Yang et al., 2025), CodeV-R1 (Zhu et al., 2025), CodeV (Zhao et al., 2025), and RTLCoder (Liu et al., 2024c), along with a traditional method, BM25 (Robertson et al., 2009), to the computation of $\bar{H}_{\text{norm}}$ (Table 2). A detailed comparison between *E203* and *Parse Lisp* is in Appendix F.

## 3.3 ChipV-RTL Overview

We now introduce **ChipV-RTL**, a novel multi-agent framework designed to automate the generation of long Verilog code from long natural language documentation (Figure 4). A complete workflow case study is provided in Appendix D.

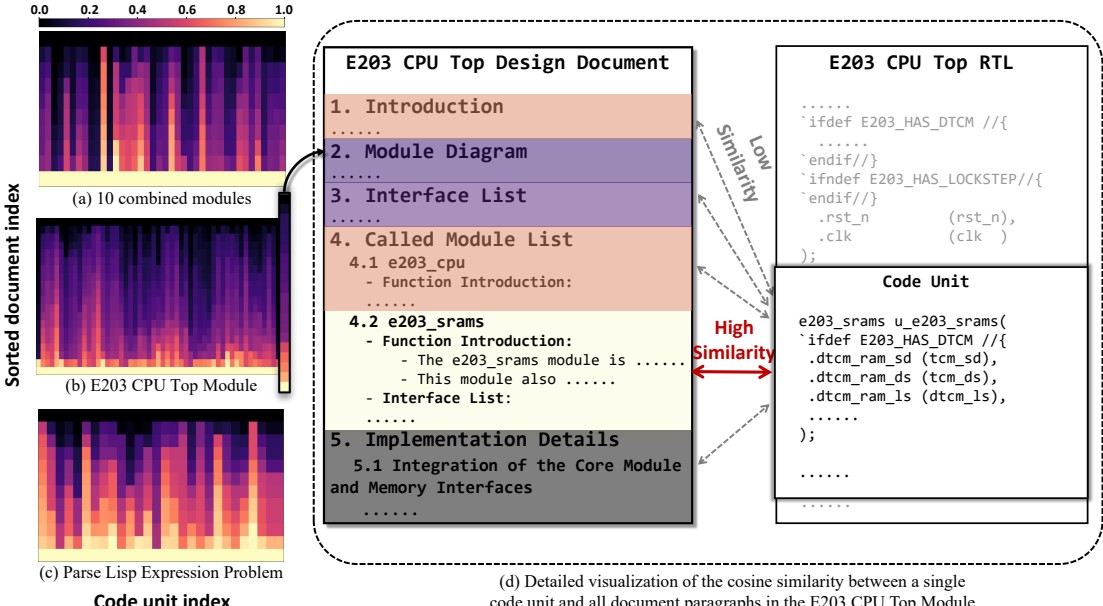

(a) 10 combined modules

(b) E203 CPU Top Module

(c) Parse Lisp Expression Problem

(d) Detailed visualization of the cosine similarity between a single code unit and all document paragraphs in the E203 CPU Top Module

*Figure 3.* **Heatmaps of normalized similarity across three tasks.** Columns represent code units, with values independently scaled [0, 1] to show cosine similarity to all document paragraphs; lower values indicate higher information locality. (a) 10 randomly selected and combined modules from VerilogEval: Extremely high locality ($\bar{H}_{\text{norm}} = 0.6366$) due to module independence. (b) E203 CPU Top Module from REALBENCH: High information locality ($\bar{H}_{\text{norm}} = 0.6718$). (c) Parse Lisp Expression problem: Typical software task with lower locality ($\bar{H}_{\text{norm}} = 0.8220$). (d) Illustration of cosine similarity between a single E203 CPU code unit and all document paragraphs.

*Table 2.* $\bar{H}_{\text{norm}}$ of Different LLMs and Methods.

| Task | BM25 | Qwen3-8B | CodeV-R1 | CodeV | RTLCoder |
|------|------|----------|----------|-------|----------|
| 10 Combined Modules | 0.5476 | 0.6366 | 0.5502 | 0.5933 | 0.6649 |
| RealBench Average | 0.6085 | 0.7261 | 0.7061 | 0.7529 | 0.7324 |
| E203 CPU Top Module | 0.6711 | 0.6718 | 0.6886 | 0.7591 | 0.6322 |
| Parse Lisp Expression | 0.8024 | 0.8220 | 0.8373 | 0.8479 | 0.7715 |

### 3.3.1 PREPROCESSING

The first stage of our pipeline structures the input documentation for efficient retrieval and comprehension by the agents. Given a raw design document, we split the text into coherent paragraphs based on its semantic section markers (e.g., "##" in Markdown). For each paragraph, an LLM is prompted to generate a dual-level description that indexes the source content, serving as keys indexing the original text segments used in subsequent stages:

**Semantic level** provides a high-level summary of the paragraph's functional intent, such as "interface specification for the DMA controller" or "timing constraints for the DDR memory interface." This supports agents with an overall understanding of the module's function.

**Lexical level** extracts fine-grained hardware-specific entities, including signal names, module identifiers, macros, and parameters, to ensure precise retrieval of low-level details that may be omitted in semantic summaries.

### 3.3.2 PLANNING AND TASK DECOMPOSITION

Using the indexed documentation from the preceding stage, the **Planner Agent** constructs the overall structure of the final Verilog code and generates a corresponding skeleton. This skeleton is expressed as pseudo-code containing syntactic placeholders that represent various code components like submodule instantiations or signal assignments.

The agent then decomposes the skeleton into sub-tasks, each corresponding to a code fragment to implement. For every sub-task, the **Retriever Agent** queries the hierarchical index to retrieve the most relevant document sections, attaching them as focused context. This granular contextualization significantly constrains the space for the next generation step, ensuring the alignment with documentation.

Unlike conventional partitioning or intermediate representations, our fragment-based decomposition minimizes overhead by mapping sub-tasks directly to a unified global design. This approach ensures tight output alignment and mitigates objective drift common in self-generated intermediate goals.

### 3.3.3 RTL GENERATION

With the sub-tasks and their associated documentation contexts prepared, multiple instances of the **RTL Agent** proceed to fill the placeholders in the code skeleton. Each agent is assigned a specific sub-task and operates within a

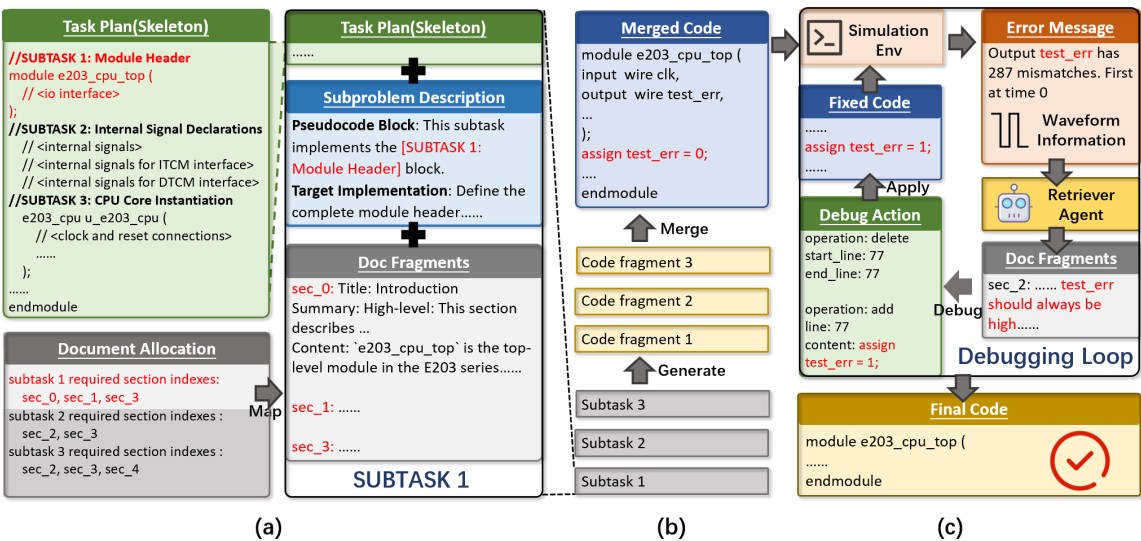

*Figure 4.* The detailed workflow of ChipV-RTL. (a) Output of the planning stage, illustrating the structure of a sub-task. (b) Overview of the code generation and merging process. (c) Overview of the debugging loop and the generation of the final code.

constrained context, allowing it to focus exclusively on its local objective. This narrow focus facilitates an accurate translation of the specification into synthesizable Verilog for the corresponding code segment, thereby reducing errors such as phantom signals and enhancing the overall quality of the generated code fragments.

### 3.3.4 CODE FRAGMENTS MERGING

After all **RTL Agents** complete fragment generation, the **Merger Agent** integrates the fragments into a correct Verilog module. To resolve potential inconsistencies or implementation errors that may arise during merging, the **Retriever Agent** first fetches relevant sections from the original documentation. Using this retrieved context, the **Merger Agent** then refines and integrates the fragments using this additional information together with the generated code, ensuring that the final output is correct and coherent.

### 3.3.5 LOCALITY-AWARE DEBUGGING

ChipV-RTL's debugging pipeline leverages **information locality** to efficiently trace errors back to their relevant documentation segments. Upon receiving the candidate Verilog code from the **Merger Agent**, we execute a simulation to obtain waveforms and error logs. To pinpoint the root cause of functional mismatches or syntax errors, we employ Abstract Syntax Tree (AST) analysis (inspired by VerilogCoder (Ho et al., 2025)). By tracing the drivers and dependency chains of the faulty signals within the AST, we identify the specific code regions and signal definitions responsible for the failure. Crucially, the **Retriever Agent** then uses this error context to fetch the small subset of documentation frag-

ments that are locally relevant to the faulty code section, as determined by the underlying information locality hypothesis. Finally, a dedicated **Debugger Agent** subsequently synthesizes the error details and the retrieved documentation to produce precise, line-number-aware edit actions (e.g., inserting or deleting specific lines). This debug loop iterates until the code is error-free or a predefined iteration limit is reached. Details are shown in Appendix A.

## 4 Experiments

We show ChipV-RTL's advantage across realistic hardware design tasks through baseline comparisons, ablation studies, and detailed analyses of on index robustness and PPA.

### 4.1 Settings

**Benchmarks.** We adopt REALBENCH (Jin et al., 2025), a challenging IP-level Verilog generation benchmark containing 60 tasks from three IPs (6 modules from AES cores, 14 modules from an SD card controller, and 40 modules from a CPU core). It features long natural language specifications (avg. 10k tokens) and complex target modules (avg. 320 lines of Verilog). To assess generalization, we also include the non-agentic spec-to-rtl subset (cid003) from CVDP (Pinckney et al., 2025). We exclude the agentic part since it involve capabilities orthogonal to RTL generation, such as reading and writing files via the command line, and navigating, organizing, and pinpointing issues across multiple code files. These requirements are beyond the topic of ChipV-RTL (IP-level spec-to-rtl).

*Table 3.* Syntax and functional pass rate comparison on the REALBENCH benchmark.

| Method | SDC Syn. | SDC Func. | AES Syn. | AES Func. | E203 CPU Syn. | E203 CPU Func. | ALL Syn. | ALL Func. |
|---|---|---|---|---|---|---|---|---|
| *Model Baselines* | | | | | | | | |
| Claude-3.7 (Anthropic, 2025) | 41.4% | 11.7% | 46.6% | 31.6% | 42.7% | 20.6% | 42.8% | 19.6% |
| DeepSeek-V3 (Liu et al., 2024a) | 44.2% | 15.3% | 55.8% | 23.3% | 19.5% | 7.5% | 28.9% | 10.9% |
| DeepSeek-R1 (Guo et al., 2025) | 49.2% | 16.4% | 66.6% | 43.3% | 11.2% | 7.6% | 25.6% | 13.2% |
| Qwen3-32B (Yang et al., 2025) | 25.3% | 15.3% | 32.4% | 16.6% | 8.3% | 6.2% | 14.7% | 9.4% |
| CodeV-R1 (Zhu et al., 2025) | 26.4% | 10.7% | 35.8% | 5.8% | 9.0% | 6.1% | 15.7% | 7.1% |
| CodeV (Zhao et al., 2025) | 2.1% | 0.0% | 1.6% | 0.0% | 1.1% | 0.2% | 1.4% | 0.1% |
| DeepRTL2 (Liu et al., 2025b) | 22.1% | 7.8% | 39.1% | 2.5% | 1.2% | 0.3% | 9.9% | 2.3% |
| VeriReason-3B (Wang et al., 2025b) | 24.6% | 7.1% | 29.1% | 1.6% | 6.2% | 3.4% | 12.8% | 4.1% |
| GPT-4o (OpenAI, 2024) | 15.7% | 5.0% | 56.6% | 5.0% | 15.1% | 5.7% | 19.4% | 5.5% |
| GPT-5 (OpenAI, 2025) | 28.2% | 13.2% | 50.0% | 43.3% | 30.1% | 12.8% | 31.6% | 16.0% |
| *Agent Baselines* | | | | | | | | |
| MAGE (Claude) (Zhao et al., 2024) | 57.1% | 21.4% | 66.6% | 33.3% | 62.5% | 20.0% | 61.6% | 21.6% |
| VerilogCoder (Claude) (Ho et al., 2025) | 50.0% | 21.4% | 66.6% | 33.3% | 27.5% | 17.5% | 36.6% | 20.0% |
| **ChipV-RTL (DeepSeek-V3)** | 64.2% | 28.5% | 50.0% | **50.0%** | 60.0% | 35.0% | 60.0% | 35.0% |
| **ChipV-RTL (Claude)** | **78.5%** | **35.7%** | **83.3%** | **50.0%** | **72.5%** | **47.5%** | **75.0%** | **45.0%** |

**Metrics.** We evaluate models on syntactic and functional correctness using each benchmark's predefined testbenches. We use pass rate (Pass@1) (OpenAI, 2021; Liu et al., 2023b) in most experiments, and extend to Pass@k in some analysis. The pass rates for direct prompting model baselines are averaged over 20 independent generations per task, whereas agent baselines are evaluated using a single generation.

**Baselines.** We establish comprehensive baselines comprising both standalone LLMs and agent-based systems. For standalone LLMs, we evaluate Claude (Claude-3.7-sonnet-250219) (Anthropic, 2025), DeepSeek-V3 (DeepSeek-v3-250324) (Liu et al., 2024a), DeepSeek-R1 (DeepSeek-r1-250528) (Guo et al., 2025), Qwen3-32B (Yang et al., 2025), CodeV-R1 (Zhu et al., 2025), CodeV (Zhao et al., 2025), GPT-4o (OpenAI, 2024), and GPT-5 (OpenAI, 2025). For agent-based approaches, we compare against SOTA methods, MAGE (Zhao et al., 2024) and VerilogCoder (Ho et al., 2025), both implemented using Claude-3.7-sonnet-250219. Our ChipV-RTL is evaluated on two different backbone models: Claude-3.7-sonnet-250219 and DeepSeek-v3-250324.

### 4.2 Main Results

Table 3 presents the evaluation results on the challenging REALBENCH benchmark. This benchmark proves particularly difficult for current LLMs, as evidenced by the modest 19.0% functional Pass@1 achieved even by the strong Claude-3.7-sonnet-250219 model. The system-level result for each module is provided in Appendix B. We have the following observations:

**ChipV-RTL achieves state-of-the-art performance with high syntactic robustness.** As shown in Table 3, ChipV-

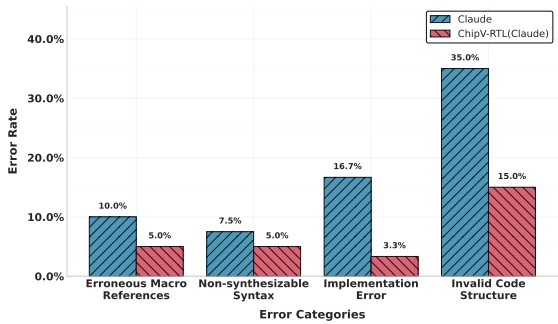

*Figure 5.* Distribution of syntactic error types for Claude 3.7 Sonnet and ChipV-RTL.

RTL significantly outperforms both monolithic models and agent-based baselines across all REALBENCH sub-tasks, achieving a 45.0% overall functional pass rate—a 23.4% absolute gain over MAGE. Also, ChipV-RTL (DeepSeek-V3) surpasses the Claude-based MAGE, demonstrating its effectiveness across different base models. This performance is further supported by the error breakdown in Fig. 5, which reveals that ChipV-RTL consistently reduces syntax errors across all categories, including hardware-specific issues like port mismatches and unsynthesizable logic (please see Appendix E for detailed failure analysis). These results underscore ChipV-RTL's ability to maintain structural integrity in complex, IP-level designs. For VerilogCoder, we adopt a reduced-cost configuration because its default workflow leads to substantially higher token consumption. Even with this cost-aware setting, VerilogCoder still consumes about 3× more tokens than ChipV-RTL, while achieving much lower functional correctness.

*Table 4.* Ablation studies on the REALBENCH benchmark.

| | SDC | | AES | | E203 CPU | | ALL | |
|---|---|---|---|---|---|---|---|---|
| **Method** | Syn. | Func. | Syn. | Func. | Syn. | Func. | Syn. | Func. |
| **ChipV-RTL** | **78.5%** | **35.7%** | 83.3% | **50.0%** | **72.5%** | **47.5%** | **75.0%** | **45.0%** |
| w/o index | 64.2% | 21.4% | **100.0%** | **50.0%** | 57.5% | 37.5% | 63.3% | 35.0% |
| w/o index & debug | 35.7% | 7.1% | 50.0% | 33.3% | 57.5% | 22.5% | 51.6% | 20.0% |
| w/o index & debug & plan | 35.7% | 7.1% | 50.0% | 33.3% | 45.0% | 22.5% | 43.3% | 20.0% |

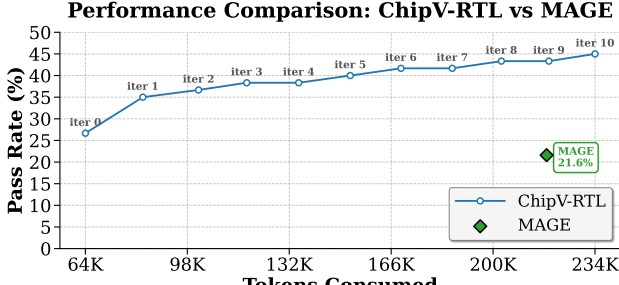

*Figure 6.* Pass@1 accuracy against cumulative token usage over ChipV-RTL's 10 debug iterations.

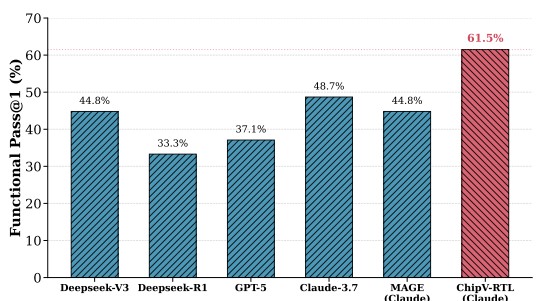

*Figure 7.* Functional pass rate comparison on CVDP cid003.

**ChipV-RTL delivers superior functional accuracy with high resource efficiency.** Beyond absolute success rates, ChipV-RTL exhibits a significantly more efficient performance-to-cost ratio. As illustrated in Fig. 6, ChipV-RTL's performance curve remains consistently above and left of MAGE's, reflecting higher functional accuracy at a fraction of the token cost. This efficiency stems from ChipV-RTL's deterministic, single-shot fragment generation strategy, which eliminates the need for the expensive, high-temperature sampling required by MAGE to reach a comparable success rate.

**Performance gains generalize across diverse benchmarks and task scales.** In addition to REALBENCH, on the CVDP (cid003) subset, ChipV-RTL significantly outperforms direct sampling and MAGE in Pass@1 (Figure 7). Although these tasks feature shorter contexts ($\sim$1,100 tokens), ChipV-RTL's strong performance demonstrates its generalization beyond long-context IP-level designs. These results confirm that ChipV-RTL maintains a competitive

*Table 5.* Retrieval quality on AES IP.

| **Backbone Model** | **Precision** | **Recall** |
|---|---|---|
| ChipV-RTL (DeepSeek-V3) | 0.93 | 0.93 |
| ChipV-RTL (Claude) | 0.92 | 0.95 |

edge across varying specification styles, underscoring its robustness regardless of task scale.

### 4.3 Ablation Studies and Further Analyses

Table 4 shows our ablation studies on ChipV-RTL:

**Indexing is indispensable for precision.** Replacing targeted fragments with full specifications overwhelms the model with irrelevant details, directly causing a 10.0% overall performance drop. This mechanism maintains focus by filtering out noise, proving that hierarchical information access is critical for managing complex IP documentation under constrained budgets.

**Debugging is non-negotiable for correctness.** Its removal, combined with the absence of indexing, reduces the pass rate to 20.0%—merely matching the base model. This indicates the debugging component ensures correctness by actively tracing errors back to specific documentation segments for targeted corrections, proving that decomposition alone cannot guarantee working IP blocks.

**The planner provides structural coherence for the generation process.** While its direct impact on functional accuracy is less pronounced, it is essential for syntactic correctness and orchestrating the multi-step task. The result of removing the planner alongside indexing and debugging yields the lowest performance, underscoring its role in maintaining a logical and executable flow for complex designs.

Additionally, we conduct further analyses:

**Robustness of Indexing.** We evaluate the robustness of our dual-level indexing through backbone model sensitivity analysis. This strategy ensures the retrieval task remains tractable and LLM-agnostic. As shown in Table 5, validation using manually annotated golden fragments from the RealBench AES IP confirms that both Claude and DeepSeek-V3 variants achieve high precision and recall ($> 0.92$), demonstrating consistent reliability across differ-

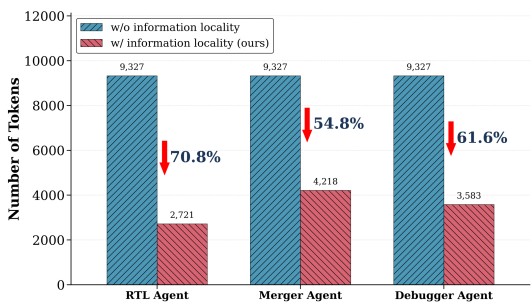

*Figure 8.* Tokens needed with and without the information locality.

ent underlying models.

**PPA Analysis.** To evaluate the hardware quality of the generated designs, we perform a PPA analysis using Yosys for logic synthesis and OpenSTA for timing and power analysis, benchmarking our results against REALBENCH. Across the synthesisable modules, the generated designs achieved improvements of 2.37% in Area, 4.34% in Delay, and 4.63% in Power compared to the Golden RTL, demonstrating that the PPA of the generated modules is highly competitive for practical hardware deployment. Details are shown in Appendix C.

**The information locality hypothesis significantly reduces the contextual overhead across various agents.** As evidenced by Figure 8, the required context was curtailed by 70.8% for the RTL Agent, 54.8% for the Merger Agent, and 61.6% for the Debugger Agent. This enhances generation accuracy while significantly lowering token consumption.

## 5 Conclusion

We present CHIPV-RTL, the first open-source framework towards IP-level design automation. Our study observes and validates the information locality of IP-level hardware specifications. Most RTL fragments can be correctly implemented based on a partial specification. Building on this insight, we design a novel hierarchical indexing strategy, a fragment-oriented task decomposition, and a locality-aware debugging loop. In REALBENCH, a real-world IP-level benchmark, ChipV-RTL delivers over 20% improvement, advancing the practical generation of reliable RTL code with LLM.

## Acknowledgment

This work is partially supported by the Strategic Priority Research Program of the Chinese Academy of Sciences (Grants No.XDB0660101, XDB0660300, XDB0660301, XDB0660302), Science and Technology Major Special Program of Jiangsu (Grant No. BG2024028), the NSF of China (Grants No.62525203, 6240073476, 62341411, U22A2028, 62502504), CAS Project for Young Scientists in Basic Research (YSBR-029) and Youth Innovation Promotion Association CAS.

## Impact Statement

This paper presents work whose goal is to advance the field of RTL code generation with LLMs. There are many potential societal consequences of our work, none of which we feel must be specifically highlighted here.

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

---

**Algorithm 1** Iterative Debugging with AST Guidance

---

1: **Input:** Verilog code $C_M$ from Merger Agent, testbench $TB$, document section descriptions $D$, max iterations $T_{max}$
2: **Output:** Verilog code after debug loop
3: $C_{curr} \leftarrow C_M$
4: $t \leftarrow 0$
5: **while** $t < T_{max}$ **do**
6:     Waveform, Errors, Pass $\leftarrow$ RunSimulation($C_{curr}, TB$)
7:     **if** Pass is **True then**
8:         **return** $C_{curr}$                                       *// Design verified successfully*
9:     **end if**
10:    WaveformInfo $\leftarrow$ TraceAST($C_{curr}$, Errors)                            *// Fault Localization via AST*
11:    *// Retrieval Step*
12:    Query $\leftarrow$ {WaveformInfo, Errors, $C_{curr}$, $D$}
13:    Docs $\leftarrow$ RetrieverAgent(Query)
14:    *// Debug Step*
15:    Prompt $\leftarrow$ {WaveformInfo, Errors, $C_{curr}$, Docs}
16:    Action $\leftarrow$ DebugAgent(Prompt)
17:    $C_{curr} \leftarrow$ ApplyEdit($C_{curr}$, Action)
18:    $t \leftarrow t + 1$
19: **end while**
20: **return** $C_{curr}$                                                  *// Return best effort if budget exhausted*

---

## A   Detailed Debugging Workflow

In this section, we provide a comprehensive description of our iterative debugging workflow. To ensure reproducibility and clarity regarding the interaction between agents, the detailed procedure is outlined in Algorithm 1.

The process begins after the Merger Agent generates a candidate Verilog code. The workflow proceeds as follows:

1. **Simulation:** We first compile and simulate the candidate code using the testbench provided by RealBench. If the simulation passes, the code is output as the final result.

2. **Fault Localization (AST-based):** Instead of feeding raw error logs directly to the LLM, we employ a Pyverilog-based AST method to trace the error signal back to its driver. This allows us to extract precise driver signals and their corresponding waveform information.

3. **Retrieval Augmented Context:** The localized AST guidance, along with error logs and the current code, is passed to the **Retriever**. The agent then queries the document descriptions to retrieve relevant reference sections.

4. **Debug Generation:** The **Debugger Agent** receives a composite prompt containing the waveform information, error logs, code context, and retrieved documents. It then generates a specific edit action to fix the identified fault.

5. **Iteration:** The edit action is applied to the Verilog code, and the cycle repeats until the testbench passes or the maximum iteration limit is reached.

## B   The System Level Result of REALBENCH

Figure 9 presents the design hierarchy of RealBench and the corresponding performance of ChipV-RTL. Specifically, it details the verification outcomes for (a) an SD card controller, (b) an AES encoder/decoder core, and (c) the Hummingbirdv2 E203 CPU Core. A "Pass" denotes successful module generation by ChipV-RTL, whereas a "Fail" indicates an unsuccessful attempt. The hierarchical tree structure within the figure visually represents the intricate task interdependencies in RealBench, underscoring its inherent complexity.

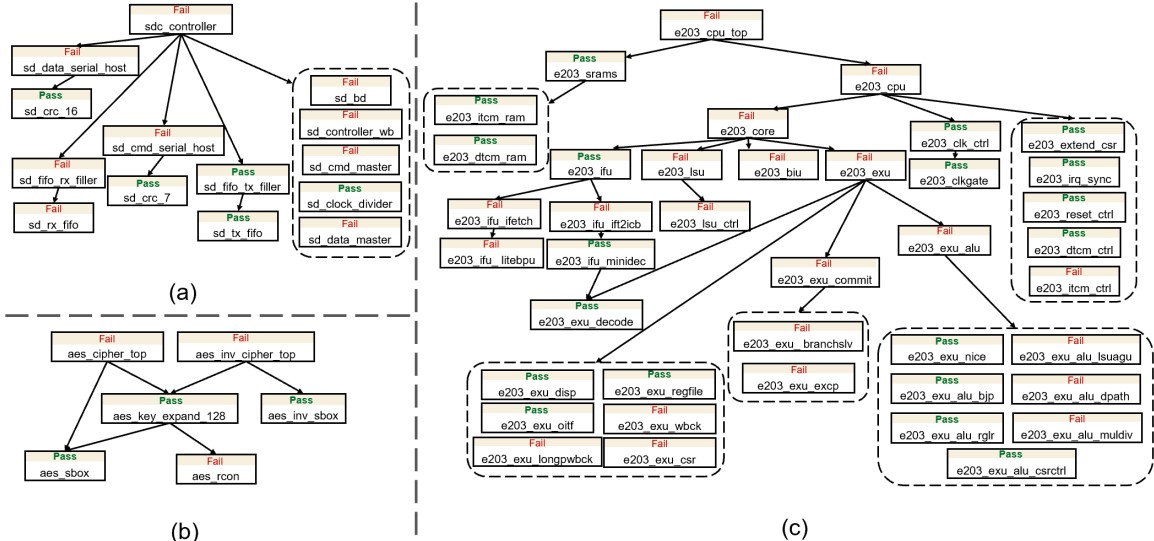

*Figure 9.* The system level result of RealBench.

## C Design Quality

To quantitatively evaluate the hardware quality of the generated designs, we conducted a comprehensive PPA (Power, Performance, and Area) analysis. We utilized Yosys for logic synthesis to obtain the area, and employed OpenSTA to report the critical path delay and total power consumption. The generated Verilog code by ChipV-RTL was benchmarked against the golden implementations sourced from the REALBENCH dataset. Table 6 presents the detailed PPA comparison for each module. Since e203_extend_csr is an empty module, its metrics are null.

It can be observed from the table that Across the synthesisable modules, the generated designs achieved improvements of 2.37% in Area, 4.34% in Delay, and 4.63% in Power compared to the Golden RTL, demonstrating that the PPA of the generated modules is highly competitive for practical hardware deployment. For some cases, the PPA metrics of the code generated by ChipV-RTL and the golden code are identical. This occurs because, although ChipV-RTL produces a different implementation from the golden code, both are synthesized into the exact same hardware structure by the synthesis tool. To show this, we calculate the similarity between the generated code and golden code, and find that after removing comments, identical module headers and syntactic elements (e.g., begin), only 14% of the code lines were identical, many of which are commonly repeated lines, such as "reg delay"; or port mappings like ".addr(addr)". We also present the code and synthesized netlist structure (Figure 10 and Figure 11) for the e203_exu_disp module. It is shown that, although their code are totally different, they share similar netlist structure and thus resulting the same PPA results.

*Table 6.* Design quality of ChipV-RTL vs. Golden RTL

| Design | Golden RTL | | | ChipV-RTL | | | Improvement | | |
|---|---|---|---|---|---|---|---|---|---|
| | Area ($\mu m^2$) | Delay (ns) | Power (mW) | Area ($\mu m^2$) | Delay (ns) | Power (mW) | Area | Delay | Power |
| aes_inv_sbox | 448.742 | 0.38 | 0.000654 | 448.742 | 0.38 | 0.000654 | 0.00% | 0.00% | 0.00% |
| aes_key_expand_128 | 3058.468 | 0.99 | 0.0193 | 2985.85 | 1.03 | 0.0177 | 2.37% | -4.04% | 8.29% |
| aes_sbox | 452.2 | 0.38 | 0.000665 | 623.77 | 0.42 | 0.000347 | -37.94% | -10.53% | 47.82% |
| e203_clk_ctrl | 31.92 | 0.52 | 5.22E-05 | 31.92 | 0.52 | 5.22E-05 | 0.00% | 0.00% | 0.00% |
| e203_clkgate | 2.128 | 0.04 | 1.18E-05 | 2.128 | 0.04 | 1.18E-05 | 0.00% | 0.00% | 0.00% |
| e203_dtcm_ctrl | 752.78 | 0.78 | 0.00125 | 758.366 | 0.78 | 0.00124 | -0.74% | 0.00% | 0.80% |
| e203_dtcm_ram | 4299995.07 | 0.14 | 99.3 | 4299995.07 | 0.14 | 99.3 | 0.00% | 0.00% | 0.00% |
| e203_exu_alu_bjp | 105.336 | 0.05 | 0.000529 | 105.336 | 0.05 | 0.000529 | 0.00% | 0.00% | 0.00% |
| e203_exu_alu_csrctrl | 138.054 | 0.31 | 0.00019 | 138.054 | 0.31 | 0.00019 | 0.00% | 0.00% | 0.00% |
| e203_exu_alu_rglr | 121.828 | 0.07 | 0.000435 | 121.828 | 0.07 | 0.000435 | 0.00% | 0.00% | 0.00% |
| e203_exu_decode | 576.688 | 0.79 | 0.000201 | 321.328 | 0.54 | 0.000132 | 44.28% | 31.65% | 34.33% |
| e203_exu_disp | 99.218 | 0.16 | 0.000158 | 99.218 | 0.16 | 0.000158 | 0.00% | 0.00% | 0.00% |
| e203_exu_nice | 156.674 | 0.37 | 0.000318 | 153.748 | 0.36 | 0.000603 | 1.87% | 2.70% | -89.62% |
| e203_exu_oitf | 743.736 | 0.45 | 0.00506 | 752.78 | 0.56 | 0.00414 | -1.22% | -24.44% | 18.18% |
| e203_exu_regfile | 8448.692 | 0.14 | 0.0939 | 8440.446 | 0.14 | 0.0939 | 0.10% | 0.00% | 0.00% |
| e203_ifu | 3106.082 | 4.58 | 0.0016 | 3106.082 | 3.83 | 0.00153 | 0.00% | 16.38% | 4.38% |
| e203_ifu_minidec | 576.688 | 0.79 | 0.000196 | 576.688 | 0.79 | 0.000196 | 0.00% | 0.00% | 0.00% |
| e203_irq_sync | 42.56 | 0.1 | 0.000371 | 42.56 | 0.1 | 0.000371 | 0.00% | 0.00% | 0.00% |
| e203_itcm_ram | 4285999.746 | 0.14 | 141 | 4285999.746 | 0.14 | 141 | 0.00% | 0.00% | 0.00% |
| e203_reset_ctrl | 12.502 | 0.1 | 0.000147 | 12.502 | 0.1 | 0.000147 | 0.00% | 0.00% | 0.00% |
| e203_srams | 8585994.816 | 0.14 | 238 | 8585995.348 | 0.14 | 208 | 0.00% | 0.00% | 12.61% |
| sd_clock_divider | 93.1 | 0.47 | 0.00061 | 93.1 | 0.47 | 0.00061 | 0.00% | 0.00% | 0.00% |
| sd_crc_16 | 128.212 | 0.23 | 0.00139 | 128.212 | 0.23 | 0.00139 | 0.00% | 0.00% | 0.00% |
| sd_crc_7 | 57.19 | 0.22 | 0.000646 | 57.19 | 0.22 | 0.000646 | 0.00% | 0.00% | 0.00% |
| sd_fifo_tx_filler | 2588.712 | 0.16 | 0.0467 | 2860.298 | 0.12 | 0.0639 | -10.49% | 25.00% | -36.83% |
| sd_tx_fifo | 2150.876 | 1.5 | 0.00554 | 1971.858 | 0.88 | 0.0072 | 8.32% | 41.33% | -29.96% |
| Average | 661380.0776 | 0.538461538 | 18.403074 | 661377.7757 | 0.481538462 | 17.24984931 | 0.25% | 3.00% | -1.15% |

---

**Golden Case: e203_exu_disp**

```
'include "e203_defines.v"
module e203_exu_disp(
  input wfi_halt_exu_req,
  output wfi_halt_exu_ack,
  input oitf_empty,
  input amo_wait,
  input disp_i_valid,
  output disp_i_ready,
  input disp_i_rs1x0,
  input disp_i_rs2x0,
  input disp_i_rs1en,
  input disp_i_rs2en,
  input ['E203_RFIDX_WIDTH-1:0] disp_i_rs1idx,
  input ['E203_RFIDX_WIDTH-1:0] disp_i_rs2idx,
  input ['E203_XLEN-1:0] disp_i_rs1,
  input ['E203_XLEN-1:0] disp_i_rs2,
  input disp_i_rdwen,
  input ['E203_RFIDX_WIDTH-1:0] disp_i_rdidx,
  input ['E203_DECINFO_WIDTH-1:0] disp_i_info,
  input ['E203_XLEN-1:0] disp_i_imm,
  input ['E203_PC_SIZE-1:0] disp_i_pc,
  input disp_i_misalgn,
  input disp_i_buserr ,
  input disp_i_ilegl ,
  output disp_o_alu_valid,
  input disp_o_alu_ready,
  input disp_o_alu_longpipe,
  output ['E203_XLEN-1:0] disp_o_alu_rs1,
  output ['E203_XLEN-1:0] disp_o_alu_rs2,
  output disp_o_alu_rdwen,
```

Golden Case: e203_exu_disp

```
output ['E203_RFIDX_WIDTH-1:0] disp_o_alu_rdidx,
output ['E203_DECINFO_WIDTH-1:0] disp_o_alu_info,
output ['E203_XLEN-1:0] disp_o_alu_imm,
output ['E203_PC_SIZE-1:0] disp_o_alu_pc,
output ['E203_ITAG_WIDTH-1:0] disp_o_alu_itag,
output disp_o_alu_misalgn,
output disp_o_alu_buserr ,
output disp_o_alu_ilegl ,
input oitfrd_match_disprs1,
input oitfrd_match_disprs2,
input oitfrd_match_disprs3,
input oitfrd_match_disprd,
input ['E203_ITAG_WIDTH-1:0] disp_oitf_ptr ,
output disp_oitf_ena,
input disp_oitf_ready,
output disp_oitf_rs1fpu,
output disp_oitf_rs2fpu,
output disp_oitf_rs3fpu,
output disp_oitf_rdfpu ,
output disp_oitf_rs1en ,
output disp_oitf_rs2en ,
output disp_oitf_rs3en ,
output disp_oitf_rdwen ,
output ['E203_RFIDX_WIDTH-1:0] disp_oitf_rs1idx,
output ['E203_RFIDX_WIDTH-1:0] disp_oitf_rs2idx,
output ['E203_RFIDX_WIDTH-1:0] disp_oitf_rs3idx,
output ['E203_RFIDX_WIDTH-1:0] disp_oitf_rdidx ,
output ['E203_PC_SIZE-1:0] disp_oitf_pc ,
input clk,
input rst_n
);
wire ['E203_DECINFO_GRP_WIDTH-1:0] disp_i_info_grp = disp_i_info ['E203_DECINFO_GRP];
wire disp_csr = (disp_i_info_grp == 'E203_DECINFO_GRP_CSR);
wire disp_alu_longp_prdt = (disp_i_info_grp == 'E203_DECINFO_GRP_AGU)
                                     ;
wire disp_alu_longp_real = disp_o_alu_longpipe;
wire disp_fence_fencei = (disp_i_info_grp == 'E203_DECINFO_GRP_BJP) &
                              ( disp_i_info ['E203_DECINFO_BJP_FENCE] | disp_i_info ['E203_DECINFO_BJP_FENCEI]);
wire disp_i_valid_pos;
wire disp_i_ready_pos = disp_o_alu_ready;
assign disp_o_alu_valid = disp_i_valid_pos;
wire raw_dep = ((oitfrd_match_disprs1) |
              (oitfrd_match_disprs2) |
              (oitfrd_match_disprs3));
wire waw_dep = (oitfrd_match_disprd);
wire dep = raw_dep | waw_dep;
assign wfi_halt_exu_ack = oitf_empty & ( amo_wait);
wire disp_condition =
             (disp_csr ? oitf_empty : 1'b1)
           & (disp_fence_fencei ? oitf_empty : 1'b1)
           & ( wfi_halt_exu_req)
           & ( dep)
           & (disp_alu_longp_prdt ? disp_oitf_ready : 1'b1);
assign disp_i_valid_pos = disp_condition & disp_i_valid;
assign disp_i_ready = disp_condition & disp_i_ready_pos;
wire ['E203_XLEN-1:0] disp_i_rs1_msked = disp_i_rs1 & {'E203_XLEN{ disp_i_rs1x0}};
wire ['E203_XLEN-1:0] disp_i_rs2_msked = disp_i_rs2 & {'E203_XLEN{ disp_i_rs2x0}};
assign disp_o_alu_rs1 = disp_i_rs1_msked;
assign disp_o_alu_rs2 = disp_i_rs2_msked;
assign disp_o_alu_rdwen = disp_i_rdwen;
assign disp_o_alu_rdidx = disp_i_rdidx;
assign disp_o_alu_info = disp_i_info;
assign disp_oitf_ena = disp_o_alu_valid & disp_o_alu_ready & disp_alu_longp_real;
```

Golden Case: e203_exu_disp

```
   assign disp_o_alu_imm = disp_i_imm;
   assign disp_o_alu_pc = disp_i_pc;
   assign disp_o_alu_itag = disp_oitf_ptr;
   assign disp_o_alu_misalgn= disp_i_misalgn;
   assign disp_o_alu_buserr = disp_i_buserr ;
   assign disp_o_alu_ilegl = disp_i_ilegl ;
   `ifndef E203_HAS_FPU
   wire disp_i_fpu = 1'b0;
   wire disp_i_fpu_rs1en = 1'b0;
   wire disp_i_fpu_rs2en = 1'b0;
   wire disp_i_fpu_rs3en = 1'b0;
   wire disp_i_fpu_rdwen = 1'b0;
   wire [`E203_RFIDX_WIDTH-1:0] disp_i_fpu_rs1idx = `E203_RFIDX_WIDTH'b0;
   wire [`E203_RFIDX_WIDTH-1:0] disp_i_fpu_rs2idx = `E203_RFIDX_WIDTH'b0;
   wire [`E203_RFIDX_WIDTH-1:0] disp_i_fpu_rs3idx = `E203_RFIDX_WIDTH'b0;
   wire [`E203_RFIDX_WIDTH-1:0] disp_i_fpu_rdidx = `E203_RFIDX_WIDTH'b0;
   wire disp_i_fpu_rs1fpu = 1'b0;
   wire disp_i_fpu_rs2fpu = 1'b0;
   wire disp_i_fpu_rs3fpu = 1'b0;
   wire disp_i_fpu_rdfpu = 1'b0;
   `endif
   assign disp_oitf_rs1fpu = disp_i_fpu ? (disp_i_fpu_rs1en & disp_i_fpu_rs1fpu) : 1'b0;
   assign disp_oitf_rs2fpu = disp_i_fpu ? (disp_i_fpu_rs2en & disp_i_fpu_rs2fpu) : 1'b0;
   assign disp_oitf_rs3fpu = disp_i_fpu ? (disp_i_fpu_rs3en & disp_i_fpu_rs3fpu) : 1'b0;
   assign disp_oitf_rdfpu = disp_i_fpu ? (disp_i_fpu_rdwen & disp_i_fpu_rdfpu ) : 1'b0;
   assign disp_oitf_rs1en = disp_i_fpu ? disp_i_fpu_rs1en : disp_i_rs1en;
   assign disp_oitf_rs2en = disp_i_fpu ? disp_i_fpu_rs2en : disp_i_rs2en;
   assign disp_oitf_rs3en = disp_i_fpu ? disp_i_fpu_rs3en : 1'b0;
   assign disp_oitf_rdwen = disp_i_fpu ? disp_i_fpu_rdwen : disp_i_rdwen;
   assign disp_oitf_rs1idx = disp_i_fpu ? disp_i_fpu_rs1idx : disp_i_rs1idx;
   assign disp_oitf_rs2idx = disp_i_fpu ? disp_i_fpu_rs2idx : disp_i_rs2idx;
   assign disp_oitf_rs3idx = disp_i_fpu ? disp_i_fpu_rs3idx : `E203_RFIDX_WIDTH'b0;
   assign disp_oitf_rdidx = disp_i_fpu ? disp_i_fpu_rdidx : disp_i_rdidx;
   assign disp_oitf_pc = disp_i_pc;
endmodule
```

ChipV-RTL Case: e203_exu_disp

```
`include "e203_defines.v"
module e203_exu_disp (
    input clk,
    input rst_n,
    input wfi_halt_exu_req,
    output wfi_halt_exu_ack,
    input oitf_empty,
    input amo_wait,
    input [`E203_ITAG_WIDTH-1:0] disp_oitf_ptr,
    output disp_oitf_ena,
    input disp_oitf_ready,
    output disp_oitf_rs1fpu,
    output disp_oitf_rs2fpu,
    output disp_oitf_rs3fpu,
    output disp_oitf_rdfpu,
    output disp_oitf_rs1en,
    output disp_oitf_rs2en,
    output disp_oitf_rs3en,
    output disp_oitf_rdwen,
    output [`E203_RFIDX_WIDTH-1:0] disp_oitf_rs1idx,
    output [`E203_RFIDX_WIDTH-1:0] disp_oitf_rs2idx,
    output [`E203_RFIDX_WIDTH-1:0] disp_oitf_rs3idx,
    output [`E203_RFIDX_WIDTH-1:0] disp_oitf_rdidx,
    output [`E203_PC_SIZE-1:0] disp_oitf_pc,
```

ChipV-RTL Case: e203_exu_disp

```
    input disp_i_valid,
    output disp_i_ready,
    input disp_i_rs1x0,
    input disp_i_rs2x0,
    input disp_i_rs1en,
    input disp_i_rs2en,
    input ['E203_RFIDX_WIDTH-1:0] disp_i_rs1idx,
    input ['E203_RFIDX_WIDTH-1:0] disp_i_rs2idx,
    input ['E203_XLEN-1:0] disp_i_rs1,
    input ['E203_XLEN-1:0] disp_i_rs2,
    input disp_i_rdwen,
    input ['E203_RFIDX_WIDTH-1:0] disp_i_rdidx,
    input ['E203_DECINFO_WIDTH-1:0] disp_i_info,
    input ['E203_XLEN-1:0] disp_i_imm,
    input ['E203_PC_SIZE-1:0] disp_i_pc,
    input disp_i_misalgn,
    input disp_i_buserr,
    input disp_i_ilegl,
    output disp_o_alu_valid,
    input disp_o_alu_ready,
    input disp_o_alu_longpipe,
    output ['E203_XLEN-1:0] disp_o_alu_rs1,
    output ['E203_XLEN-1:0] disp_o_alu_rs2,
    output disp_o_alu_rdwen,
    output ['E203_RFIDX_WIDTH-1:0] disp_o_alu_rdidx,
    output ['E203_DECINFO_WIDTH-1:0] disp_o_alu_info,
    output ['E203_XLEN-1:0] disp_o_alu_imm,
    output ['E203_PC_SIZE-1:0] disp_o_alu_pc,
    output ['E203_ITAG_WIDTH-1:0] disp_o_alu_itag,
    output disp_o_alu_misalgn,
    output disp_o_alu_buserr,
    output disp_o_alu_ilegl,
    input oitfrd_match_disprs1,
    input oitfrd_match_disprs2,
    input oitfrd_match_disprs3,
    input oitfrd_match_disprd
);
    wire disp_csr = (disp_i_info['E203_DECINFO_GRP] == 'E203_DECINFO_GRP_CSR);
    wire disp_agu = (disp_i_info['E203_DECINFO_GRP] == 'E203_DECINFO_GRP_AGU);
    wire disp_bjp = (disp_i_info['E203_DECINFO_GRP] == 'E203_DECINFO_GRP_BJP);
    wire disp_fence = disp_bjp & disp_i_info['E203_DECINFO_BJP_FENCE];
    wire disp_fencei = disp_bjp & disp_i_info['E203_DECINFO_BJP_FENCEI];
    wire need_wait_oitf_empty = disp_csr | disp_fence | disp_fencei;
    wire rs1_dep_oitf_idx = oitfrd_match_disprs1;
    wire rs2_dep_oitf_idx = oitfrd_match_disprs2;
    wire rs3_dep_oitf_idx = oitfrd_match_disprs3;
    wire disp_raw_dep = rs1_dep_oitf_idx | rs2_dep_oitf_idx | rs3_dep_oitf_idx;
    wire disp_waw_dep = oitfrd_match_disprd;
    wire ['E203_XLEN-1:0] disp_i_rs1_msked = disp_i_rs1x0 ? {'E203_XLEN{1'b0}} : disp_i_rs1;
    wire ['E203_XLEN-1:0] disp_i_rs2_msked = disp_i_rs2x0 ? {'E203_XLEN{1'b0}} : disp_i_rs2;
    wire oitf_empty_condition = ( need_wait_oitf_empty) | (need_wait_oitf_empty & oitf_empty);
    wire no_dep_condition = ( disp_raw_dep) & ( disp_waw_dep);
    wire wfi_halt_condition = wfi_halt_exu_req;
    wire agu_oitf_ready = ( disp_agu) | (disp_agu & disp_oitf_ready);
    wire disp_condition = oitf_empty_condition & wfi_halt_condition & no_dep_condition & agu_oitf_ready;
    assign disp_i_ready = disp_condition & disp_o_alu_ready;
    assign disp_oitf_ena = disp_o_alu_longpipe & disp_o_alu_valid & disp_o_alu_ready;
    assign disp_o_alu_valid = disp_condition & disp_i_valid;
    assign disp_o_alu_rs1 = disp_i_rs1_msked;
    assign disp_o_alu_rs2 = disp_i_rs2_msked;
    assign disp_o_alu_rdwen = disp_i_rdwen;
    assign disp_o_alu_rdidx = disp_i_rdidx;
```

ChipV-RTL Case: e203_exu_disp

```
    assign disp_o_alu_info = disp_i_info;
    assign disp_o_alu_imm = disp_i_imm;
    assign disp_o_alu_pc = disp_i_pc;
    assign disp_o_alu_itag = disp_oitf_ptr;
    assign disp_o_alu_misalgn = disp_i_misalgn;
    assign disp_o_alu_buserr = disp_i_buserr;
    assign disp_o_alu_ilegl = disp_i_ilegl;
    assign disp_oitf_rs1en = disp_i_rs1en;
    assign disp_oitf_rs2en = disp_i_rs2en;
    assign disp_oitf_rdwen = disp_i_rdwen;
    assign disp_oitf_rs1idx = disp_i_rs1idx;
    assign disp_oitf_rs2idx = disp_i_rs2idx;
    assign disp_oitf_rdidx = disp_i_rdidx;
    assign disp_oitf_pc = disp_i_pc;
    assign disp_oitf_rs3en = 1'b0;
    assign disp_oitf_rs3idx = {'E203_RFIDX_WIDTH{1'b0}};
'ifdef E203_HAS_FPU
    assign disp_oitf_rs1fpu = 1'b0;
    assign disp_oitf_rs2fpu = 1'b0;
    assign disp_oitf_rs3fpu = 1'b0;
    assign disp_oitf_rdfpu = 1'b0;
'else
    assign disp_oitf_rs1fpu = 1'b0;
    assign disp_oitf_rs2fpu = 1'b0;
    assign disp_oitf_rs3fpu = 1'b0;
    assign disp_oitf_rdfpu = 1'b0;
'endif
    assign wfi_halt_exu_ack = oitf_empty & ( amo_wait);
endmodule
```

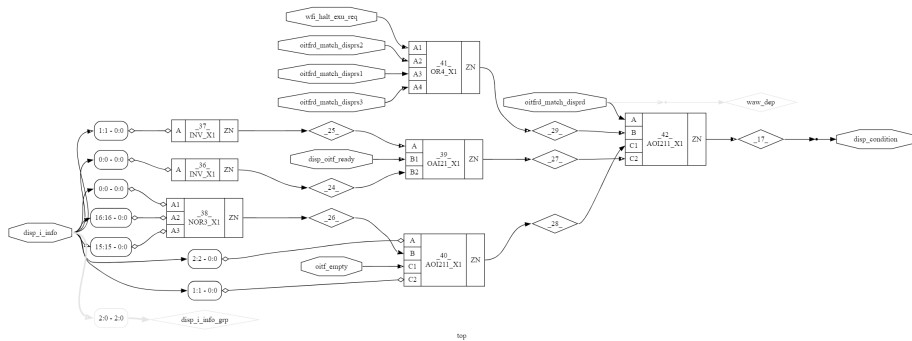

*Figure 10.* The netlist of the substructure of golden e203_exu_disp module after synthesis

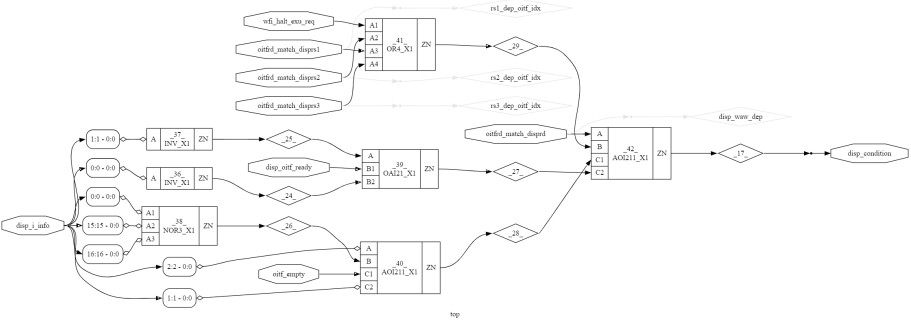

*Figure 11.* The netlist of the substructure of ChipV-RTL's e203_exu_disp module after synthesis

# D  Intermediate Results of ChipV-RTL

To better illustrate ChipV-RTL's workflow, this section delves into the detailed intermediate results for the **e203_exu** problem in REALBENCH. We'll display the outputs generated by ChipV-RTL agents, including document fragments, pseudocode, plans, code fragments, and debug actions, providing a comprehensive understanding of the process.

---

**Document Fragments**

"section_0": "The 'e203_exu' module represents the execution unit (EXU) of a processor core, responsible for executing instructions received from the instruction fetch unit (IFU) and handling exceptions, pipeline flushes, and write-back operations. This module integrates submodules for decoding, dispatching, ALU operations, commit handling, and more, ensuring proper execution of instructions and system state management.",
"section_1": "| Direction | Port Name | Width | Description |

| ———— | ————- | ————- | ———————————————— |
| output | commit_mret | 1 | Indicates that an MRET instruction has been committed. |
| output | commit_trap | 1 | Indicates that a trap (e.g., exception) has been committed. |
| output | exu_active | 1 | Indicates whether the execution unit is currently active. |
| output | excp_active | 1 | Indicates whether an exception is currently active. |
| output | core_wfi | 1 | Indicates that the core is in a waiting-for-interrupt (WFI) state. |
| output | tm_stop | 1 | Indicates that the timer should stop. |
| output | itcm_nohold | 1 | Indicates no hold on the instruction TCM. |
| output | core_cgstop | 1 | Core clock gating stop signal. |
| output | tcm_cgstop | 1 | TCM clock gating stop signal. |
| input | core_mhartid | E203_HART_ID_W | Hardware thread ID of the core. |
| input | dbg_irq_r | 1 | Debug interrupt request signal. |
......",
......

---

Pseudocode

```
// === SUBTASK 1: Module Header and IO Port Definitions ===
module e203_exu (
// <basic interface signals>
// <debug control interface signals>
// <IFU IR stage interface signals>
// <flush interface signals>
// <LSU write-back interface signals>
// <AGU ICB interface signals>
// <optional CSR_NICE interface signals>
// <optional NICE interface signals>
);
// === SUBTASK 2: Internal Signal Declarations ===
// <register file interface signals>
// <decode stage signals>
// <dispatch stage signals>
// <OITF signals>
// <ALU signals>
// <commit stage signals>
// <CSR signals>
// <long-pipeline write-back signals>
// <write-back signals>
// === SUBTASK 3: Submodule Instantiations ===
// Register File
e203_exu_regfile u_e203_exu_regfile (
// <register file connections>
);
// Decode
e203_exu_decode u_e203_exu_decode (
// <decode connections>
);
// Dispatch
e203_exu_disp u_e203_exu_disp (
// <dispatch connections>
);
// OITF
e203_exu_oitf u_e203_exu_oitf (
// <OITF connections>
);
// ALU
e203_exu_alu u_e203_exu_alu (
// <ALU connections>
);
// Commit
e203_exu_commit u_e203_exu_commit (
// <commit connections>
);
// CSR
e203_exu_csr u_e203_exu_csr (
// <CSR connections>
);
// Long-pipeline Write-back
e203_exu_longpwbck u_e203_exu_longpwbck (
// <long-pipeline write-back connections>
);
// Write-back
e203_exu_wbck u_e203_exu_wbck (
// <write-back connections>
);
// === SUBTASK 4: Conditional Logic and Signal Assignments ===
// <exu_active signal logic>
// <dependency information logic>
// <optional NICE interface logic>
// <optional CSR_NICE interface logic>
endmodule
```

---

Plan

"subproblems": [
{ "id": "module_header_and_io", "description": "**Pseudocode Block**: This subtask implements the [SUBTASK 1: Module Header and IO Port Definitions] block in the provided pseudocode. **Target Implementation**: Define the complete module header with all interface signals including basic interface, debug control interface, IFU IR stage interface, flush interface, LSU write-back interface, AGU ICB interface, and optional CSR_NICE and NICE interfaces according to the specifications.", "required_section_indexes": [ "section_0", "section_1", "section_2", "section_3", "section_4", "section_5", "section_6", "section_7", "section_8" ] },
{ "id": "internal_signals", "description": "**Pseudocode Block**: This subtask implements the [SUBTASK 2: Internal Signal Declarations] block in the provided pseudocode. **Target Implementation**: Define all internal signals needed for connecting the submodules, including register file interface signals, decode stage signals, dispatch stage signals, OITF signals, ALU signals, commit stage signals, CSR signals, long-pipeline write-back signals, and write-back signals.", "required_section_indexes": [ "section_9", "section_10", "section_11", "section_12", "section_13", "section_14", "section_15", "section_16", "section_17" ] },
...... ]

---

Code Fragment

```
// === SUBTASK 4: Conditional Logic and Signal Assignments ===
// exu_active signal logic
// Active when OITF is not empty, valid instruction entering, or exception active
assign exu_active = (~ oitf_empty) | i_valid | excp_active;
// dependency information logic for IFU
// dec2ifu_rden: Indicates whether current instruction writes to an integer register
assign dec2ifu_rden = dec_rdwen & (~dec_rdwen_fp);
// dec2ifu_rs1en: Indicates whether current instruction reads rs1 integer register
assign dec2ifu_rs1en = dec_rs1en & (~dec_rs1en_fp);
// dec2ifu_rdidx: Destination register index
assign dec2ifu_rdidx = dec_rdidx;
// rf2ifu_rs1: Value of rs1 register
assign rf2ifu_rs1 = read_src1_dat;
`ifdef E203_HAS_NICE
// optional NICE interface logic
assign nice_cmt_off_ilgl_o = nice_cmt_off_ilgl_i;
assign nice_xs_off = nice_xs_off_i;
`endif
`ifdef E203_HAS_CSR_NICE
// optional CSR_NICE interface logic
assign csr_nice_csr_valid = csr_nice_csr_valid_i;
assign csr_nice_csr_ready = csr_nice_csr_ready_i;
assign csr_nice_csr_addr = csr_nice_csr_addr_i;
assign csr_nice_csr_wr = csr_nice_csr_wr_i;
assign csr_nice_csr_wdata = csr_nice_csr_wdata_i;
assign csr_nice_csr_rdata = csr_nice_csr_rdata_i;
`endif
```

---

Debug Action

"fix_operations": [ { "operation": "delete_block", "start_line": 838, "end_line": 843 }, { "operation": "add_block", "line": 838, "content": [ " // Flush interface logic - handle different flush conditions", " // Flush can be triggered by exceptions, interrupts, or control flow instructions", " assign pipe_flush_req = commit_trap | commit_mret;", " ", " // For normal exceptions and interrupts, use the trap vector", " // For returns from exceptions (mret), use the saved EPC", " assign pipe_flush_add_op1 = commit_mret ? csr_epc_r : csr_mtvec_r;", " assign pipe_flush_add_op2 = 32'b0;", "`ifdef E203_TIMING_BOOST", " assign pipe_flush_pc = commit_mret ? csr_epc_r : csr_mtvec_r;", "`endif" ] }, { "operation": "delete_block", "start_line": 817, "end_line": 819 }, { "operation": "add_block", "line": 817, "content": [ " // exu_active signal logic", " // Active when OITF is not empty, valid instruction entering, or exception active", " // Also active during flush operations", " assign exu_active = (~oitf_empty) | i_valid | excp_active | pipe_flush_req;" ] } ],

---

# E  Failure Cases

Although ChipV-RTL achieves significant performance gains over the baseline, it still faces persistent challenges in RealBench. These failure cases can be broadly categorized into three types: complex logic, syntactic errors, and excessive signals. In this section, we present examples of these three types of failure. Among these instances, the aes_cipher_top module failed to realize the complex encryption logic required. The e203_itcm_ctrl module encountered errors due to the

improper application of macro expressions to define bit widths. In the case of e203_exu_alu_csrctrl, the failure was caused by incorrect use of comma triggered by an ifdef directive. Lastly, the e203_cpu_top module failed because the model confused signal directionality because of the excessive volume of signals.

---

**Failure Case: complex logic**

```
module aes_cipher_top(
......
);
......
// Instantiate 16 S-boxes for SubBytes transformation
aes_sbox u_sbox_00 (.a(sa00_r), .b(sa00_out));
......
// Instantiate key expansion module
aes_key_expand_128 u_key_expand (
......
);
......
// Combinational logic - state machine and transformations
always @(*) begin
......
case (state_r)
IDLE: begin
done_next = 1'b0;
if (ld) begin
state_next = INIT_ROUND;
sa00_next = text_in[127:120]; sa10_next = text_in[119:112];
......
INIT_ROUND: begin
......
ROUND_OP: begin
......
......
endmodule
```

---

**Failure Case: syntactic errors**

```
'include "e203_defines.v"
module e203_itcm_ctrl (
......
);
......
assign sram_icb_cmd_wdata = sram_sel_ifu ? ('E203_ITCM_DATA_WIDTH-'E203_XLEN)'b0, ifu2itcm_icb_cmd_wdata :
arbt_icb_cmd_wdata;
assign sram_icb_cmd_wmask = sram_sel_ifu ? ('E203_ITCM_WMSK_WIDTH-'E203_XLEN/8)'b0, ifu2itcm_icb_cmd_wmask :
arbt_icb_cmd_wmask;
assign sram_icb_cmd_size = sram_sel_ifu ? 2'b10 : arbt_icb_cmd_size; // IFU always uses word access
// Connect response signals from SRAM controller
assign ifu2itcm_icb_rsp_valid = sram_sel_ifu & sram_icb_rsp_valid;
assign arbt_icb_rsp_valid = sram_sel_arbt & sram_icb_rsp_valid;
assign sram_icb_rsp_ready = (sram_sel_ifu & ifu2itcm_icb_rsp_ready) |
(sram_sel_arbt & arbt_icb_rsp_ready);
......
endmodule
```

---

**Failure Case: syntactic errors**

```
'include "e203_defines.v"
module e203_exu_alu_csrctrl (
......
// Clock and reset
input wire clk,
input wire rst_n,
// NICE interface signals
'ifdef E203_HAS_CSR_NICE
// NICE interface signals
......
output wire [31:0] nice_csr_wdata,
input wire [31:0] nice_csr_rdata
'endif
);
......
endmodule
```

---

**Failure Case: excessive signals**

```
module e203_cpu_top (
......
// PPI ICB interface
input wire ppi_icb_cmd_valid,
output wire ppi_icb_cmd_ready,
input wire ['E203_ADDR_SIZE-1:0] ppi_icb_cmd_addr,
input wire ppi_icb_cmd_read,
......
);
......
e203_cpu u_e203_cpu (
// Clock and reset connections
.clk (clk),
.rst_n (rst_n),
......
// PPI ICB interface connections
.ppi_icb_enable (ppi_icb_enable),
.ppi_icb_cmd_valid (ppi_icb_cmd_valid),
.ppi_icb_cmd_ready (ppi_icb_cmd_ready),
.ppi_icb_cmd_addr (ppi_icb_cmd_addr),
.ppi_icb_cmd_read (ppi_icb_cmd_read),
......
)
......
endmodule
```

## F   Information Locality Case

In this section, we present e203_srams as an instance of high information locality, while contrasting it with Parse Lisp Expression, which exhibits low information locality.

The e203_srams specification from RealBench exhibits clear modularity. The functional description and interface definitions of each task (ITCM RAM and DTCM RAM) are grouped tightly together in dedicated sections. This allows sub-tasks to be implemented using only partial, relevant documentation without interference from other sub-modules.

Conversely, the Parse Lisp Expression task demonstrates weaker information locality. The description is broad, making it hard to pinpoint specific paragraphs that correspond to the code's abstract algorithms. For example, the stack structure required for implementation has no corresponding section in the document; instead, it must be abstracted from the overall problem description. As a result, the full document is usually required to understand and design the complete algorithm.

Good Case: e203_srams document

# e203_srams Design Documentation
## 1. Introduction
The e203_srams module is the memory management module of the E203 processor. It is mainly used for integrating and managing the Instruction Tightly Coupled Memory (ITCM) and Data Tightly Coupled Memory (DTCM). This module flexibly controls the instantiation of ITCM and DTCM through macro definitions 'E203_HAS_ITCM' and 'E203_HAS_DTCM'.
## 2. Module Block Diagram

## 3. Interface Definition
### General Interface
| Signal Name | Direction | Bit Width | Description |
|———|———|———|———|
| test_mode | Input | 1 | Unused and unassigned |

**Good Case: e203_srams document**

### ITCM RAM Interface Signals exist only if the 'E203_HAS_ITCM' is defined

| Signal Name | Direction | Bit Width | Description |
|——|——|——|——|
| itcm_ram_sd | Input | 1 | ITCM power off enable signal |
| itcm_ram_ds | Input | 1 | ITCM deep sleep mode enable |
| itcm_ram_ls | Input | 1 | ITCM light sleep mode enable |
| itcm_ram_cs | Input | 1 | ITCM chip select signal |
| itcm_ram_we | Input | 1 | ITCM write enable signal |
| itcm_ram_addr | Input | E203_ITCM_RAM_AW | ITCM address |
| itcm_ram_wem | Input | E203_ITCM_RAM_MW | ITCM write mask |
| itcm_ram_din | Input | E203_ITCM_RAM_DW | ITCM write data |
| itcm_ram_dout | Output | E203_ITCM_RAM_DW | ITCM read data |
| clk_itcm_ram | Input | 1 | ITCM clock signal |
| rst_itcm | Input | 1 | ITCM reset signal |

### DTCM RAM Interface

Signals exist only if the 'E203_HAS_DTCM' is defined
(Similar to the ITCM interface, with the signal name prefix changed to dtcm).

| Signal Name | Direction | Bit Width | Description |
|——|——|——|——|
| dtcm_ram_sd | Input | 1 | DTCM power off enable signal |
| dtcm_ram_ds | Input | 1 | DTCM deep sleep mode enable |
| dtcm_ram_ls | Input | 1 | DTCM light sleep mode enable |
| dtcm_ram_cs | Input | 1 | DTCM chip select signal |
| dtcm_ram_we | Input | 1 | DTCM write enable signal |
| dtcm_ram_addr | Input | E203_ITCM_RAM_AW | DTCM address |
| dtcm_ram_wem | Input | E203_ITCM_RAM_MW | DTCM write mask |
| dtcm_ram_din | Input | E203_ITCM_RAM_DW | DTCM write data |
| dtcm_ram_dout | Output | E203_ITCM_RAM_DW | DTCM read data |
| clk_dtcm_ram | Input | 1 | DTCM clock signal |
| rst_dtcm | Input | 1 | DTCM reset signal |

## 4. Submodule List

### ITCM RAM

#### Function

The e203_dtcm_ram module is a Data Tightly Coupled Memory (DTCM) RAM module for the E203 processor. The module is encapsulated based on a generic RAM module, primarily used for data storage and access. The module is controlled by the macro definition 'E203_HAS_DTCM'.

#### Interface

| Signal Name | Direction | Width | Description |
|——|——|——|——|
| sd | Input | 1 | Power domain shutdown enable signal for power management |
| ds | Input | 1 | Deep sleep mode enable, controlling complete power area shutdown |
| ls | Input | 1 | Light sleep mode enable, reducing power without full shutdown |
| cs | Input | 1 | Chip select signal, controlling RAM selection |
| we | Input | 1 | Write enable signal, controlling write operation |
| addr | Input | E203_ITCM_RAM_AW | Address input, specifying read/write location |
| wem | Input | E203_ITCM_RAM_MW | Write mask, controlling specific byte writing |
| din | Input | E203_ITCM_RAM_DW | Data input to be written |
| rst_n | Input | 1 | Asynchronous reset signal (active low) |
| clk | Input | 1 | System clock |
| dout | Output | E203_ITCM_RAM_DW | Data output, read data |

### DTCM RAM

#### Function

The e203_itcm_ram module is an Instruction Tightly Coupled Memory (ITCM) RAM module for the E203 processor. The module is encapsulated based on a generic RAM module, primarily used for instruction storage and access. The module is controlled by the macro definition 'E203_HAS_ITCM'.

#### Interface

| Signal Name | Direction | Width | Description |
|——|——|——|——|
| sd | Input | 1 | Power domain shutdown enable signal for power management |
| ds | Input | 1 | Deep sleep mode enable, controlling complete power area shutdown |
| ls | Input | 1 | Light sleep mode enable, reducing power without full shutdown |
| cs | Input | 1 | Chip select signal, controlling RAM selection |

---

**Good Case: e203_srams document**

| we | Input | 1 | Write enable signal, controlling write operation |
| addr | Input | E203_DTCM_RAM_AW | Address input, specifying read/write location |
| wem | Input | E203_DTCM_RAM_MW | Write mask, controlling specific byte writing |
| din | Input | E203_DTCM_RAM_DW | Data input to be written |
| rst_n | Input | 1 | Asynchronous reset signal (active low) |
| clk | Input | 1 | System clock |
| dout | Output | E203_DTCM_RAM_DW | Data output, read data |
## 5. Implementation Details
1. Memory management mechanism
- Supports independent configuration and control of ITCM and DTCM.
- Each memory module has an independent clock and reset domain.
2. Data flow control
- Adopts a preprocessed data output mechanism (dout_pre).
- Removes the data bypass function in test mode.
3. Submodule Instantiation Details
The submodule interface is connected to the corresponding interface of this module. For example, the 'sd' signal of
'e203_itcm_ram' is connected to the 'itcm_ram_sd' interface.
## 6. Limitations
1. Functional constraints
- The address must be within the valid range.

---

**Good Case: e203_srams code**

```
'include "e203_defines.v"
module e203_srams(
    ......
);
'ifdef E203_HAS_ITCM //
wire ['E203_ITCM_RAM_DW-1:0] itcm_ram_dout_pre;
e203_itcm_ram u_e203_itcm_ram (
    .sd (itcm_ram_sd),
    .ds (itcm_ram_ds),
    .ls (itcm_ram_ls),
    .cs (itcm_ram_cs ),
    .we (itcm_ram_we ),
    .addr (itcm_ram_addr ),
    .wem (itcm_ram_wem ),
    .din (itcm_ram_din ),
    .dout (itcm_ram_dout_pre ),
    .rst_n(rst_itcm ),
    .clk (clk_itcm_ram )
    );
     assign itcm_ram_dout = itcm_ram_dout_pre;
'endif//
'ifdef E203_HAS_DTCM //
wire ['E203_DTCM_RAM_DW-1:0] dtcm_ram_dout_pre;
e203_dtcm_ram u_e203_dtcm_ram (
    .sd (dtcm_ram_sd),
    .ds (dtcm_ram_ds),
    .ls (dtcm_ram_ls),
    .cs (dtcm_ram_cs ),
    .we (dtcm_ram_we ),
    .addr (dtcm_ram_addr ),
    .wem (dtcm_ram_wem ),
    .din (dtcm_ram_din ),
    .dout (dtcm_ram_dout_pre ),
    .rst_n(rst_dtcm ),
    .clk (clk_dtcm_ram )
    );
     assign dtcm_ram_dout = dtcm_ram_dout_pre;
'endif//
endmodule
```

### Bad Case: Parse Lisp Expression document

You are given a string expression representing a Lisp-like expression to return the integer value of.

The syntax for these expressions is given as follows.

An expression is either an integer, let expression, add expression, mult expression, or an assigned variable. Expressions always evaluate to a single integer. (An integer could be positive or negative.)

A let expression takes the form "(let v1 e1 v2 e2 ... vn en expr)", where let is always the string "let", then there are one or more pairs of alternating variables and expressions, meaning that the first variable v1 is assigned the value of the expression e1, the second variable v2 is assigned the value of the expression e2, and so on sequentially; and then the value of this let expression is the value of the expression expr.

An add expression takes the form "(add e1 e2)" where add is always the string "add", there are always two expressions e1, e2 and the result is the addition of the evaluation of e1 and the evaluation of e2.

A mult expression takes the form "(mult e1 e2)" where mult is always the string "mult", there are always two expressions e1, e2 and the result is the multiplication of the evaluation of e1 and the evaluation of e2.

For this question, we will use a smaller subset of variable names. A variable starts with a lowercase letter, then zero or more lowercase letters or digits. Additionally, for your convenience, the names "add", "let", and "mult" are protected and will never be used as variable names.

Finally, there is the concept of scope. When an expression of a variable name is evaluated, within the context of that evaluation, the innermost scope (in terms of parentheses) is checked first for the value of that variable, and then outer scopes are checked sequentially. It is guaranteed that every expression is legal. Please see the examples for more details on the scope.

Example 1:

Input: expression = "(let x 2 (mult x (let x 3 y 4 (add x y))))" Output: 14

Explanation: In the expression (add x y), when checking for the value of the variable x, we check from the innermost scope to the outermost in the context of the variable we are trying to evaluate. Since x = 3 is found first, the value of x is 3.

Example 2:

Input: expression = "(let x 3 x 2 x)"

Output: 2

Explanation: Assignment in let statements is processed sequentially.

Example 3:

Input: expression = "(let x 1 y 2 x (add x y) (add x y))"

Output: 5

Explanation: The first (add x y) evaluates as 3, and is assigned to x. The second (add x y) evaluates as 3+2 = 5.

Constraints:

1 ¡= expression.length ¡= 2000

There are no leading or trailing spaces in expression. All tokens are separated by a single space in expression. The answer and all intermediate calculations of that answer are guaranteed to fit in a 32-bit integer. The expression is guaranteed to be legal and evaluate to an integer.

Bad Case: Parse Lisp Expression code

```
def implicit_scope(func):
    def wrapper(*args):
        args[0].scope.append()
        ans = func(*args)
        args[0].scope.pop()
        return ans
    return wrapper
class Solution(object):
    def __init__(self):
        self.scope = []
    @implicit_scope
    def evaluate(self, expression):
        if not expression.startswith('('):
            if expression[0].isdigit() or expression[0] == '-':
                return int(expression)
            for local in reversed(self.scope):
                if expression in local: return local[expression]
        tokens = list(self.parse(expression[5 + (expression[1] == 'm'): -1]))
        if expression.startswith('(add'):
            return self.evaluate(tokens[0]) + self.evaluate(tokens[1])
        elif expression.startswith('(mult'):
            return self.evaluate(tokens[0]) * self.evaluate(tokens[1])
        else:
            for j in xrange(1, len(tokens), 2):
                self.scope[-1][tokens[j-1]] = self.evaluate(tokens[j])
            return self.evaluate(tokens[-1])
    def parse(self, expression):
        bal = 0
        buf = []
        for token in expression.split():
            bal += token.count('(') - token.count(')')
            buf.append(token)
            if bal == 0:
                yield " ".join(buf)
                buf = []
        if buf:
            yield " ".join(buf)
```

## G   Experimental Settings

We provide the parameter settings used in our experiments. All methods are evaluated under the RealBench. For open-source models, we use the same prompt template and inference backend across all evaluated tasks.

In addition, the settings of agent-based baselines are chosen with token-budget considerations. For MAGE, we set the candidate parameters so that its overall token budget is comparable to ChipV-RTL. For VerilogCoder, its default configuration incurs substantially higher token consumption. Therefore, we reduce the listed round and auto-reply parameters to approximately one-tenth of their default values while keeping the original workflow unchanged.

| Parameter | Value |
|---|---|
| ChipV-RTL: Temperature | 0.1 |
| ChipV-RTL: Top-$p$ | 1.0 |
| ChipV-RTL: Retry limits | 2 |
| open-source models: Temperature | 0.6 |
| open-source models: Top-$p$ | 1.0 |
| open-source models: Backend | VLLM 0.8.5 |
| MAGE: rtl_max_candidates | 2 |
| MAGE: rtl_selected_candidates | 1 |
| MAGE: sim_max_retry | 2 |
| VerilogCoder: plan_graph_retrieval group_chat max_round | 10 |
| VerilogCoder: verilog_completion group_chat max_round | 10 |
| VerilogCoder: verilog_waveform_debug group_chat max_round | 4 |
| VerilogCoder: verilog_debug group_chat max_round | 4 |
| VerilogCoder: plan_graph_retrieval max_consecutive_auto_reply | 2 |
| VerilogCoder: verilog_completion max_consecutive_auto_reply | 2 |
| VerilogCoder: verilog_waveform_debug max_consecutive_auto_reply | 4 |
| VerilogCoder: verilog_debug max_consecutive_auto_reply | 4 |
| VerilogCoder: verilog_completion HistoryLimit max_messages | 1 |
| VerilogCoder: verilog_waveform_debug HistoryLimit max_messages | 1 |

*Table 7.* Experimental settings in RealBench.

