# OpenReview forum: "QiMeng-ChipV-RTL: Exploiting Information Locality for IP-level Verilog Generation"
_ICML.cc/2026/Conference — ICML 2026 regular_

### Official Review · Reviewer_Smon · 2026-02-13

**Soundness:** 3
**Presentation:** 3
**Significance:** 3
**Originality:** 3
**Overall Recommendation:** 4
**Confidence:** 4

**Summary:**

This paper proposes a approach to generate IP-level verilog code. It outputs a module consisting of several semantic code units instead of a monolithic text file. The authors state their information locality assumption and quantify it by measuring the entropy. Based on this, authors design a comprehensive framework to solve the IP-level verilog code generation task, which splits the document to several sections and retrieves based on the entropy. Experiments on RealBench demonstrate both better performance and lower token cost.

**Compliance With Llm Reviewing Policy:**

Affirmed.

**Final Justification:**

I appreciate the detailed rebuttal from the authors. They clearly answered my questions regarding the VerilogCoder baseline token consumption and explained how the framework can still access the full specification when needed. The core idea of exploiting information locality is well motivated and the empirical results show genuine improvements in both success rate and efficiency. Taking the rebuttal and other reviews into account I still find this to be a technically solid paper with a clear contribution to RTL generation. I am keeping my score as a weak accept.

**Key Questions For Authors:**

1. (Please refer to weakness 2) Is the sub-agent or the full framework capable to access the full document?
2. The workflow appears to be strictly sequential from planning to generation. If the planner agent makes mistakes, does the framework have the opportunity to fix the bug? Is there a feedback mechanism that allows the system to re-plan or adjust the skeleton?
3. Could the authors provide the runtime comparison with other baselines?

**Limitations:**

Limitations (failure cases) are discussed in Appendix E

**Strengths And Weaknesses:**

Strengths
1. Both the assumption and the definition of the information locality are well demonstrated and with clear motivation.
2. The proposed framework is easy to understand and fully utilizes the proposed the information locality assumption.
3. Both performance and token cost are better than other baselines.

Weaknesses
1. In table 3, all pass rate of VerilogCoder (Claude) is 0%. The authors said that it fail completely due to the excessive token costs and low completion rates. It could be better to give a clearer count of its token consumption.
2. Under some cases, if the full document (or many different sections which are distant in the original document) is essential for module design, is the sub-agent still capable to address this question? Or is the sub-agent/full-framework accessible to the full document?

---

> ### Author Rebuttal · Authors · 2026-03-31
>
> Thank you for your valuable feedback. Your main concerns include the behavior of VerilogCoder, whether the framework can access the global context, the replanning mechanism, and runtime efficiency.
>
> In response, we have provided a detailed analysis for VerilogCoder, clarified that our retriever has access to the full specification (including distant sections), explained the built-in correction mechanisms across stages, and added runtime comparisons with baselines. We will revise our manuscript accordingly.
>
> >**W1: VerilogCoder results**
>
> Thank you for pointing this out. We have conducted additional experiments in the rebuttal to better understand this result.
> The 0.0% score of VerilogCoder on RealBench is mainly due to its extremely high token cost under a comparable budget setting.
>
> We tested a highly reduced version (limiting most stages from 100 to 10 rounds), which still consumed 40.7M tokens (~3$\times$ our full LocalV pipeline) and achieved only 20.0% accuracy. We will clarify this analysis in the revised paper.
>
> >**W2: Access to distributed context**
>
> Thank you for raising this question. The retriever has access to the entire specification through the hierarchical index and is not limited to a single contiguous chunk. It can retrieve multiple relevant sections, even if they are far apart in the original document.
>
> >**Q1: Full document accessibility**
>
> Thank you for your question. We have explained in W2.
>
> >**Q2: Error correction mechanism**
>
> Thank you for raising this question. The workflow is not strictly one-pass. Even if the planner makes mistakes, later stages can correct them.
> In particular, the merger agent can revise or adjust the planner's structure, and the debug agent further improves robustness by identifying and fixing errors.
> Moreover, all stages are grounded in retrieved fragments from the original specification, which helps reduce error propagation in practice.
>
> >**Q3: Runtime comparison**
>
> Thank you for the suggestion. We report the average runtime, number of model calls and tokens on RealBench:
>
> |Method|Runtime|API calls|Tokens|
> |-|-|-|-|
> |LocalV|617.3s|19.0|224k|
> |VerilogCoder|2458.7s|885.8|678k|
> |MAGE|1043.3s|9.3|218k|
>
> LocalV is significantly faster with an acceptable token cost in practice, mainly because its RTL generation stage on different fragments can be parallelized.
>
> ---
> Thank you again for your valuable feedback. We hope the rebuttal is adequate in addressing your concerns, and if so, please update your review in accordance. If any questions remain, we welcome further discussion.

---

> > ### Author Rebuttal · Reviewer_Smon · 2026-04-02
> >
> > I thank the authors for their rebuttal. My evaluation remains consistent with my initial 'Weak Accept' rating after taking the ongoing discussion and other reviewers' points into account. I will sustain my original score.

---

> > > ### Author Response · Authors · 2026-04-05
> > >
> > > Dear reviewer Smon,
> > >
> > > We sincerely thank you for your valuable feedback! If our rebuttal has solved all of your concerns as you noted, could you please reconsider your score like reviewer p5sb and jig1?
> > >
> > > Also, we are limited to reply only once for each reviewer, but we are happy to adopt any further suggestions from you. Thank you again!

---

### Official Review · Reviewer_jig1 · 2026-02-25

**Soundness:** 4
**Presentation:** 3
**Significance:** 3
**Originality:** 4
**Overall Recommendation:** 5
**Confidence:** 4

**Summary:**

This paper proposes a locality-driven multi-agent framework for generating IP-level Verilog from long specifications. The key idea is to decompose long-document, long-code generation into localized sub-tasks using document indexing, retrieval, structured planning, and fragment merging. The system is evaluated on REALBENCH and CVDP and shows clear improvements over prior agent-based approaches on long-context RTL generation tasks.

The work is technically solid and practically relevant for LLM-based hardware design. The main contribution lies in the pipeline for handling long specifications and large code outputs rather than in individual agent components. While a few details (reproducibility settings, baseline behavior, and positioning) could be clarified more, the approach is well motivated and likely useful for future work in this area.

**Compliance With Llm Reviewing Policy:**

Affirmed.

**Final Justification:**

I'm suggesting acceptance for this paper.

The original paper's soundness and originality is already excellent, and the author's rebuttal has addressed my questions and suggestion properly, leading to a predictable improvement on paper's presentation. Therefore I increased my score and kept my positive suggestion.

**Key Questions For Authors:**

See weaknesses.

**Limitations:**

Yes. The paper includes a reasonable discussion of limitations and potential societal impact. The main remaining limitation is discussed in weakness part.

**Strengths And Weaknesses:**

**Soundness.** The paper is technically solid overall. The method is clearly described and evaluated on two relevant benchmarks (REALBENCH and CVDP), with comparisons to both strong standalone models and prior agent-based systems. The ablations are useful and support the claim that document indexing, planning, and merging are important components.

There are a few minor soundness and reproducibility issues. Some decoding and sampling settings (e.g., temperature, top-p, retry limits) are not specified, which makes reproduction harder and should be included in an appendix. In addition, the VerilogCoder baseline yields near-zero results, which is somewhat surprising. The paper briefly notes token and completion issues, but a clearer explanation of the failure modes would help readers interpret the comparison.

**Presentation.** The paper is clearly written and easy to follow. The pipeline structure and workflow are well illustrated. The locality metric and system design are described in a way that an expert reader can understand and reproduce with additional configuration details.

The related work section could be expanded slightly. In particular, recent work that builds structured internal representations (“mental models”) of hardware designs for generation is relevant (like AssertionForge or Cadence ChipStack) and could be discussed to clarify how this work differs. Also, the claim that hardware and software differ in locality is supported by limited software-side evidence (essentially one example of “Parse Lisp Expression”), which makes the cross-domain claim somewhat narrow. Additionally, there are also a few minor presentation issues: Figure 2 and Table 1 describe slightly different agent structures (the Retriever Agent appears in Table 1 but not in Figure 2), and Table 4 has formatting inconsistencies where bolded numbers do not always correspond to the best values in each column. These are small issues but should be corrected for clarity.

**Significance.** The paper addresses a relevant problem: scaling LLM-based RTL generation to longer specifications and larger designs. The proposed pipeline for handling long context and long code is likely to be useful for future work in LLM-based EDA systems and long-context code generation more broadly. Even though the impact is mostly within the hardware design domain, the practical value is clear.

Part of the motivation relies on long-context degradation shown using earlier-generation models. Since newer models have much larger context windows, it would be helpful for the paper to clarify how the proposed approach scales with newer models.

**Originality.** The main contribution is the locality-driven pipeline for long-document to long-code generation, including document indexing, retrieval, planning, and merging. Some individual components (multi-agent loops, AST-based debugging) are built on prior work such as VerilogCoder and MAGE, but the integration of these ideas into a coherent pipeline for IP-level RTL generation is a meaningful contribution. The entropy-based locality metric also supports the motivation beyond traditional BM25. Overall, the work represents a solid system-level contribution that others in the area are likely to build on.

---

> ### Author Rebuttal · Authors · 2026-03-31
>
> Thank you for your valuable feedback. Your main concerns include missing experimental details, unclear baseline behaviors, limited related work discussion, and some presentation issues.
>
> In response, we have provided decoding settings and code for reproducibility, added detailed analysis of VerilogCoder, expanded related work, and software-side experiments. We will revise our manuscript accordingly.
>
> >**Soundness W1: Some decoding and sampling settings are not specified.**
>
> Thank you for pointing this out. We will include all decoding and sampling settings in the appendix to ensure reproducibility. We have also provided our code in the Supplementary Material.
>
> For clarity, the main settings used in LocalV are:
>
> |Parameter|Value|
> |-|-|
> |Temperature |0.1|
> |Top-p|1.0|
> |Retry limits|2|
>
> The temperature and top-p are default values without any tuning, and they are the same as the parameters in VerilogCoder. We will include a complete list of parameters in the revised paper.
>
> >**Soundness W2: The VerilogCoder baseline yields near-zero results, which is somewhat surprising.**
>
> Thank you for pointing this out. We have conducted additional experiments in the rebuttal to better understand this result.
>
> The 0.0% score of VerilogCoder on RealBench is mainly due to its extremely high token cost under a comparable budget setting.
> We tested a highly reduced version (limiting most stages from 100 to 10 rounds), which still consumed 40.7M tokens (~3$\times$ our full LocalV pipeline) and achieved only 20.0% accuracy. We will clarify this analysis in the revised paper.
>
> >**Presentation W1: The related work section could be expanded slightly.**
>
> Thank you for the suggestion. We will expand the related work section and clarify the differences.
>
> Methods such as AssertLLM and AssertionForge mainly target verification/assertion generation and build global structured representations (e.g., graphs) from both the specification and the design under test. Similarly, ChipStack constructs structured internal representations ("mental models") to organize design information for generation without many method details disclosed.
>
> In contrast, LocalV targets a different task, i.e., IP-level RTL generation with long-context and long code. Additionally, LocalV exploits information locality to retrieve only relevant fragments for each sub-task, which avoids global reconstruction and thus becomes more efficient.
>
> >**Presentation W2: The claim that hardware and software differ in locality is supported by limited software-side evidence.**
>
> Thank you for the suggestion. We agree that the hardware-vs-software locality claim should be stated more carefully, and we will revise the wording to avoid overgeneralization.
>
> We have also expanded the software-side evidence by adding four long LeetCode hard problems. Across these five problems, the average entropy scores (BM25, Qwen3-8B, CodeV-R1, CodeV, RTLCoder) are 0.8191, 0.8300, 0.8292, 0.8276, and 0.8030, respectively, which are consistently higher than in the hardware setting.
>
> |task|BM25|Qwen3-8B|CodeV-R1|CodeV|RTLCoder|
> |-|-|-|-|-|-|
> |Parse Lisp Expression|0.8024|0.8360|0.8373|0.8479|0.8171|
> |Cracking the Safe|0.8770|0.8129|0.7850|0.7940|0.7527|
> |Race Car|0.7948|0.8211|0.8186|0.8349|0.7947|
> |Tag Validator|0.7907|0.8200|0.8512|0.8357|0.8120|
> |Cut Off Trees for Golf Event|0.8308|0.8603|0.8543|0.8259|0.8388|
> |Average|0.8191|0.8300|0.8292|0.8276|0.8030|
>
> >**Presentation W3: There are also a few minor presentation issues.**
>
> Thank you for pointing this out. We will correct these presentation issues for clarity.
>
> >**Significance W1: Part of the motivation relies on long-context degradation shown using earlier-generation models.**
>
> Thank you for your concern. Longer context windows alone do not resolve the problem. As shown in Table 3, we evaluate strong modern long-context models such as Claude 3.7 Sonnet (200k tokens) and GPT-5 (400k tokens), whose context is large enough to include all documents, code, and even the full LocalV input/output. However, they still perform significantly worse than LocalV on RealBench.
>
> LocalV can help these strong models with its structured decomposition, targeted retrieval, and verification. These core components can be directly combined with newer models and are supposed to further improve their performance.
>
> ---
> Thank you again for your valuable feedback. We hope the rebuttal is adequate in addressing your concerns, and if so, please update your review in accordance. If any questions remain, we welcome further discussion.

---

> > ### Author Rebuttal · Reviewer_jig1 · 2026-04-03
> >
> > Thanks the author for the additional explanation and experiments.
> >
> > Especially, per the Presentation W2 experiment, I would still encourage a evaluation on a larger SW dataset/benchmark (like size at ~100 level) instead of hand-picking 5 examples; But I understand that the rebuttal time is short and the paper itself should be good given the authors are already considering to revise the wording to avoid overgeneralization.
> >
> > In general I'm suggesting this paper for acceptance and therefore increasing my score.

---

> > > ### Author Response · Authors · 2026-04-05
> > >
> > > Dear reviewer jig1,
> > >
> > > We sincerely thank you for your positive feedback! We are glad that our rebuttal has addressed your concerns. We appreciate your insightful suggestions and will incorporate them into the final manuscript.

---

### Official Review · Reviewer_dCYW · 2026-03-11

**Soundness:** 3
**Presentation:** 3
**Significance:** 2
**Originality:** 2
**Overall Recommendation:** 4
**Confidence:** 4

**Summary:**

This paper proposes LocalV, a multi-agent framework for IP-level Verilog generation from long natural language specifications. The core idea is an information locality hypothesis: many RTL code units can be generated from only a localized subset of the specification. Based on that premise, the method combines dual-level document indexing, task planning, localized RTL generation, fragment merging, and AST-guided debugging. The reported results are strong on the chosen benchmarks.

**Compliance With Llm Reviewing Policy:**

Affirmed.

**Final Justification:**

Based on the rebuttal, I'm increasing my score.

**Key Questions For Authors:**

1. Can the authors clarify what the core ML novelty is beyond extending prior agentic RTL generation frameworks to long-context hardware specifications?

2. How does the post-hoc entropy-based locality analysis, which relies on ground-truth code units, validate the effectiveness of the actual deployed retrieval system?

3. Since BM25 outperforms some LLM-based methods in Table 2, how should readers interpret the claimed advantage of semantic/model-based locality over simple lexical matching?

4. VerilogCoder achieves 0.0% on REALBENCH, which is highly unusual, so can the authors provide a detailed explanation of its setup and failure modes?

5. Can the authors provide a more complete efficiency analysis?

**Limitations:**

Yes

**Strengths And Weaknesses:**

Paper Strengths:

1. The paper targets a genuinely hard setting, long specifications, long outputs, which is closer to realistic EDA workflows than small academic benchmarks, which is well-motivated.

2. The multi-agent design follows human intuition, including planning, retrieval, localized generation, merging, and AST-guided debugging.

3. The empirical improvements are substantial on the main benchmark.

4. It is also great attempt to formalize the intuition behind the method via the locality metric rather than presenting the approach as pure engineering. The entropy-based analysis is a useful descriptive addition.


Weaknesses

1. The ML novelty is limited. I could just view this work as an extension of the previous RTL generation agentic framework on long-context RTL generation. It will be a great work for conference like DAC or ICCAD, but perhaps under the level of ICML.

2. The locality analysis weakly supports the core claims. The post-hoc entropy metric relies on ground-truth data, disconnecting it from the actual LLM retrieval system.

3. In Table 2, BM25 scores better than some LLMs. This suggests the effect comes from basic word matching, rather than a unique ability of the model.

4. VerilogCoder scoring 0.0% on REALBENCH is highly unusual and requires detailed explanation and diagnosis in the main text.

5. The cost analysis is largely token-based and does not give a truly complete accounting of latency, number of model calls

6. The system is given a simulation environment and benchmark testbenches. That is reasonable for controlled evaluation, but for practical design automation this is a powerful external oracle.

---

> ### Author Rebuttal · Authors · 2026-03-31
>
> Thank you for the valuable feedback. Your main concerns include (1) ML novelty, (2) the validity of locality analysis, (3) evaluation details, and (4) the use of testbenches.
>
> We have clarified our core contribution on IP-level generation and locality hypothesis,  added new experiments, and explained evaluation settings. We will revise our manuscript accordingly.
>
> >**W1 & Q1: Limited ML novelty**
>
> Thanks for the concern. LocalV's novelty lies in the discovery and formalization of the **Information Locality**, moving beyond incremental extensions of existing RTL agents. While prior works focus on simple tasks, we target **more realistic and substantially harder IP-level generation** (e.g., 241.2 vs. 56.1 code lines, 197.3 vs. 22.3 doc lines, and 520 vs. 48.1 cells).
> Specifically, our novelty is in three aspects:
>
> Conceptually, we identify and quantitatively validate information locality in IP-level RTL generation: many RTL fragments are primarily determined by a localized subset of the specification (shown in Fig. 3 and Table 2).
>
> Algorithmically, we turn this into a new generation framework with hierarchical indexing, fragment-level decomposition, localized generation, and locality-aware debugging, which factorizes not only what to generate, but also what context is actually needed for each subproblem.
>
> Empirically, we show its effectiveness on realistic hardware tasks with much longer specifications and code, where LocalV substantially outperforms both prior RTL-generation agentic frameworks (MAGE and VerilogCoder) and strong LLMs like GPT-5 on RealBench and CVDP.
>
> We will revise the paper to make this ML contribution and its distinction from prior agentic RTL frameworks clearer.
>
> >**W2 & Q2: Weak locality validation**
>
> Thanks for the concern. The entropy metric serves as an **oracle-style, theoretical motivation analysis**, which diagnoses whether IP-level RTL generation exhibits locality and does not truly affect LocalV's performance.
>
> For the exploitation of this property, we designed LocalV's indexing and retrieval mechanism. We evaluated them in Table 5 (against annotated golden fragments), Table 4, and Fig. 8, showing that these mechanism improves both performance and context efficiency.
>
> >**W3 & Q3: Lexical vs semantic effects**
>
> Thanks for the concern. Table 2 shows how strongly different methods "perceive" the locality property, so the absolute scores in Table 2 are not directly comparable across methods since different methods produce different score distributions. What matters is the relative comparison across tasks within the same method.
>
> The purpose of Table 2 is to show that different approximation methods, including lexical and semantic ones, consistently reveal the locality property in IP-level Verilog generation.
>
> LocalV uses both lexical and semantic information through dual-level indexing, so its improvements stem from the full locality-aware pipeline rather than simple word matching alone.
>
> >**W4 & Q4: VerilogCoder results**
>
> Thanks for the concern. We have conducted additional experiments for a better understanding.
>
> The 0.0% score of VerilogCoder on RealBench is due to its extremely high token cost under a comparable budget setting.
> Specifically, we tested a highly reduced version (limiting stages from 100 to 10 rounds), which still consumed 40.7M tokens (~3$\times$ our LocalV pipeline) and achieved only 20.0% accuracy. We will clarify this in the revised paper.
>
> >**W5 & Q5: Efficiency analysis**
>
> Thanks for the suggestion. We will report the average runtime, number of model calls and tokens on RealBench:
>
> |Method|Runtime|API calls|Tokens|
> |-|-|-|-|
> |LocalV|617.3s|19.0|224k|
> |VerilogCoder|2458.7s|885.8|678k|
> |MAGE|1043.3s|9.3|218k|
>
> LocalV is significantly faster with an acceptable token cost in practice, mainly because its RTL generation stage on different fragments can be parallelized.
>
> >**W6: Reliance on testbench**
>
> Thanks for the concern. In professional hardware development, design and verification (DV) are decoupled into separate teams that work independently based on the same design specification. The testbench essentially acts as the golden reference that the RTL must satisfy; if a mismatch occurs, both teams collaborate to resolve ambiguities in the specification or implementation. Our approach mimics this industry reality, where a designer uses the verification feedback from testbenches to iteratively refine their logic. By treating the verification environment as a given component, we can isolate and evaluate the agent's specific capability in RTL generation, which is the core focus of this work. This setting is also consistent with existing SOTA agentic methods like MAGE and VerilogCoder, ensuring a fair and reproducible comparison.
>
> ---
> Thank you again for your valuable feedback. We hope the rebuttal is adequate in addressing your concerns, and if so, please update your review in accordance. If any questions remain, we welcome further discussion.

---

> > ### Author Rebuttal · Reviewer_dCYW · 2026-04-05
> >
> > I appreciate your rebuttal but I am not convinced that information locality is a core ML contribution but I do appreciate that you are handling larger more complex designs and that information locality makes sense.  I'm also not sure it makes sense to test VeriCoder with less tokens than that which it was published, but it is clear that LocalV requries fewer tokens.

---

> > > ### Author Response · Authors · 2026-04-07
> > >
> > > Dear Reviewer dCYW,
> > >
> > > We sincerely thank you for your positive feedback! We understand your concern on the ML contribution, but we kindly believe that our insights on challenging and realistic ML applications (i.e., IP-level RTL generation) are also meaningful to ML, as they reveal useful structural properties and design principles for ML-based systems. We will revise the paper to clarify this point.
> > >
> > > Regarding VerilogCoder, we appreciate your comment on the evaluation setting. We will clarify that its performance is evaluated under a comparable token budget, as the original setting incurs extremely high token cost, which is difficult for us to sustain in practice -- in particular, even a reduced version already incurs extremely high token cost (e.g., 40.7M tokens) with limited accuracy. We will revise the paper to present this setup and its implications more clearly.

---

### Official Review · Reviewer_p5sb · 2026-03-13

**Soundness:** 3
**Presentation:** 3
**Significance:** 3
**Originality:** 3
**Overall Recommendation:** 5
**Confidence:** 3

**Summary:**

This paper introduces an agentic framework for verilog code generation from specifications. It analyses the challenges of current frameworks and finds, that the specification/code length is a big issue. To mitigate authors aim to divide the problems into smaller subproblems based on the intuition that information is inherently local to modules. To verify this intuition, authors come up with a locality measure, that quantifies how distinct the information of a given component is compared to other available information. After verifying the claim, authors move on to propose a preprocessing step that divides the problem into subproblems. Subsequently agents operate on the subproblems that are merged, simulated and debugged using the information specific to that locality, allowing to debug with better specific context awareness.
The results suggest that this enables higher success rate than prior work with fewer tokens on the investigated benchmarks.

**Compliance With Llm Reviewing Policy:**

Affirmed.

**Final Justification:**

The rebuttal addressed the questions I have raised. Therefore, I believe this is interesting work I recommened to accept (5). I increased my score accordingly.

**Key Questions For Authors:**

- What happens if you use thinking models/enable thinking mode on the models used (like Qwen3 or GPT5)?
- How does your approach compare to high performance LLMs when they are fine-tuned for verilog generation tasks?
- Why is the comparison limited to two related approaches (Spec2RTL, RTLSquad) and maybe one of the reinforcement learning ones for direct comparisons?
- The results in Table 2 seem to suggest high variance between LLMs in the locality measure. Is there a direct correlation between the performance of your framework and the locality measure?

**Limitations:**

No limitations are discussed in the main content. What are the complexity limitations it can handle right now? Are there specific designs/tasks it struggles with? e.g. are there tasks with low locality where it falls behind?

**Strengths And Weaknesses:**

# Strengths And Weaknesses (Soundness, Presentation, Significance, Originality)

## Soundness
### Strengths
- Ablation studies about tokens used and syntactic error rates make a compelling case for the method.
- Failure cases in the supplement provide basic understanding about what things may go wrong.

### Major Weaknesses
- While the main table compares to two other agentic approaches and many llms, the reinforcement category is completely omitted and the choice of other agentic frameworks is not sufficiently explained.
- The locality measure seems to show clear and significant differences between different examples. However, the locality strength varies a lot depending on the LLM used. Moreover, when considering the "baseline" for the 10 combined modules is between 54 and 66 and the max for the LeetCode is between 77 and 85, the range of the locality measure is not very large. Considering this range, RealBench often sits right in the middle of it, therefore being a mixture of both local and global importance. However, the specific locality does not seem to be considered in the framework itself, even though it is introduced at length.

### Minor Weaknesses
- "these [LLM-based RTL generation] models remain primarily effective for small-scale tasks and often falter when applied to complex, IP-level specifications." (l.104) As these approaches are never compared to in this work, so claiming they are bad without referencing other studies seems  like an over claim.
- It is unclear to me why the comparison in Tab.1 has not been done to Spec2RTL and RTLSquad as well.
- The first contribution about identifying challenges is overly broad. From the subsequent analysis, the issue of complex debugging does not seem to be a new problem as both other agentic frameworks already use an agent to handle this. The debugging just seem to become easier due to the breakdown into smaller parts, which also makes every other component (other than merging) easier.

## Presentation
### Strengths
Presentation quality is very high generally speaking.
- Well written, the specific challenges present today are clearly explained.
- Figures are laid out well and support understanding of the presented framework.
- Reproducibility: reproducing results based on the provided framework descriptions might be challenging but code is there allowing to dive in deeper if necessary.

### Weaknesses
- Small mistakes/typos
  - l.67: "We discover the information locality hypothesis ..." I don't think one discovers a hypothesis, I recommend replacing the word.
  - l.306: "Finally, A dedicated "; a should not be capitalized
  - l.308: "whereas Agent baselines"; agent should not be capitalized

## Significance
Automating design can greatly benefit chip development. The consideration of locality in the design seems to have clear benefits to the generation. Moreover, the way the tasks is divided into subparts and handled by the retriever/merger agents seems to be applicable to other works in the area too. However, as the specific design seems to matter a lot, providing more detailed descriptions (instructions) of all agents would be required to generalize the findings to other works.

## Originiality
Originality is good. While other papers have done agent based approaches in the past, the paper leverages locality measurements to improve token efficiency and final results compared to prior works.

---

> ### Author Rebuttal · Authors · 2026-03-31
>
> Thanks for the valuable feedback. Your main concerns include (1) more comparisons (e.g., RL and prior RTL methods), (2) the validity and role of the locality measurement, (3) clarification on contributions (especially debugging) and settings.
>
> In response, we have added new comprehensive experiments and clarified the role of locality measurement, and we will revise the paper for better clarity.
>
> > **Soundness W1 & Q3: Comparison with RL**
>
> Thanks for the concern. In Table 3, we selected MAGE and VerilogCoder as representative baselines, as they are the strongest reproducible RTL generation agents available at the time of submission.
>
> We already evaluated several RL-trained models in Table 3, including CodeV-R1 and DeepSeek-R1. We have also added VeriReason (see Soundness W3), which performs worse than LocalV on Realbench.
>
> > **Soundness W2: Locality measurement**
>
> Thanks for the concern. (1) Variation across LLMs is expected. Measuring locality requires approximating underlying probabilities, but current LLMs are not trained for this and cannot estimate them perfectly like an oracle, so different models produce different results. (2) The "10 combined modules" is a synthetic lower bound created by concatenating unrelated tasks, which is unreal. It is therefore natural for RealBench to fall between this lower bound and software tasks like LeetCode. (3) Locality is used by our framework in two ways: each subtask is paired with retrieved local document fragments during planning and generation, and AST-based fault localization retrieves only the relevant parts of the document during debugging.
>
> > **Soundness W3: RTL model comparisons**
>
> Thanks for the concern. We agree that the original wording may sound too strong. We will revise it to be precise.
> We have also evaluated several new representative open-source methods mentioned:
>
> |Method|Accuracy|
> |-|-|
> |CodeV|0.1%|
> |CodeV-R1|7.1%|
> |RTLCoder|3.4%|
> |DeepRTL2|2.3%|
> |VeriReason-3B|4.1%|
>
> These results remain consistent with our claim.
>
> > **Soundness W4: Spec2RTL/RTLSquad comparisons**
>
> Thanks for the concern. We focused on two representative and open-source methods in Table 1 to align with experiments.
> Spec2RTL requires human intervention and reconstructs context for each subtask, which may lose information, while LocalV is fully automatic and directly retrieves relevant parts of the original specification. RTLSquad targets relatively simple problems and mainly uses an iterative loop without explicit task decomposition, whereas LocalV performs a locality-aware task decomposition.
>
> We will include them in the revised version. However, because open-source implementations were unavailable at the time of submission, we cannot perform evaluations on them.
>
> > **Soundness W5: Contribution on debugging**
>
> Thanks for the concern. We agree that the wording may be misleading. We do not intend to claim that debugging was first identified as a challenge, but rather to highlight debugging in the setting of long documents and complex IP-level code, which has been less explored. We will revise this to be more precise.
>
> > **Presentation W1 & Significance W1: Minor issues and details**
>
> Thanks for the advice. We will fix these issues and add details like prompts and instructions.
>
> > **Q1: Effect of thinking models**
>
> Thanks for the concern. In fact, the results of Qwen3 and GPT-5 were already in thinking mode (GPT-5 is medium thinking).
> Following your advice, we have added more results (QW for Qwen3):
>
> |Method|Accuracy|
> |-|-|
> |QW-8B no think|4.4%|
> |QW-8B think|4.5%|
> |QW-32B no think|6.4%|
> |QW-32B think|9.4%|
> |GPT-5 min.|14.5%|
> |GPT-5 med.|16.0%|
> |GPT-5 high|14.4%|
>
> Overall, thinking does not lead to a large improvement.
>
> > **Q2: Comparison with fine-tuned models**
>
> Thanks for the concern. We compared with fine-tuned models such as CodeV, which perform worse than LocalV.
> For strong LLMs like GPT-5, fine-tuning them is computationally expensive, and such fine-tuning requires high-quality IP-level data, which remains challenging.
>
> We also note that LocalV is orthogonal to fine-tuning and can be combined with it for further gains.
>
> > **Q4: Locality-performance correlation**
>
> Thanks for the concern. There are 3 different concepts: the locality measure, the intrinsic locality of the document, and the performance of our framework. The locality measure itself serves as a motivation analysis only and is not part of our method, so its variance across LLMs does not affect our framework.
>
> > **L1: Limitation analysis**
>
> Thanks for the concern. Our failures mainly fall into 3 categories: complex logic, syntactic errors, and excessive signals (Appendix E). We will revise our paper to make it clear.
>
> ---
> Thank you again! We hope our rebuttal adequately addresses your concerns and kindly ask the updating of your review. We welcome further discussion.

---

> > ### Author Rebuttal · Reviewer_p5sb · 2026-04-01
> >
> > Thank you for your additional insights, clarifications and experiments. While I don’t think all points should start with a „Thanks for the concern“ (which is nice but a little redundant), overall, the rebuttal addresses the questions I have raised. Hence, I decided to increase the score.

---

> > > ### Author Response · Authors · 2026-04-05
> > >
> > > Dear reviewer p5sb,
> > >
> > > We sincerely thank you for your positive feedback! We are glad that our rebuttal has addressed your concerns. We appreciate your insightful suggestions and will incorporate them into the final manuscript.

---

### Decision · Program_Chairs · 2026-04-30

**Decision:**

Accept (regular)

**Comment:**

This paper addresses an important and practical problem in IP-level Verilog generation from long specifications. Reviewers generally agreed that the proposed locality-driven decomposition framework is technically solid, well motivated, and effective, with clear gains in both success rate and efficiency on realistic benchmarks. The rebuttal further strengthened the paper by clarifying baseline behavior, adding comparisons and runtime analysis, and improving the discussion of locality and reproducibility. While some reviewers noted that the contribution is stronger as a systems advance than as a fundamentally new ML method, the overall evaluation is clearly positive. I therefore recommend accept.